# Distinct signals in medial and lateral VTA dopamine neurons modulate fear extinction at different times

Lili X Cai[1], Katherine Pizano[1], Gregory W Gundersen[2], Cameron L Hayes[1], Weston T Fleming[1], Sebastian Holt[1], Julia M Cox[1], Ilana B Witten[1,3]*

[1]Princeton Neuroscience Institute, Princeton University, Princeton, United States;
[2]Department of Computer Science, Princeton University, Princeton, United States;
[3]Department of Psychology, Princeton University, Princeton, United States

**Abstract** Dopamine (DA) neurons are thought to encode reward prediction error (RPE), in addition to other signals, such as salience. While RPE is known to support learning, the role of salience in learning remains less clear. To address this, we recorded and manipulated VTA DA neurons in mice during fear extinction. We applied deep learning to classify mouse freezing behavior, eliminating the need for human scoring. Our fiber photometry recordings showed DA neurons in medial and lateral VTA have distinct activity profiles during fear extinction: medial VTA activity more closely reflected RPE, while lateral VTA activity more closely reflected a salience-like signal. Optogenetic inhibition of DA neurons in either region slowed fear extinction, with the relevant time period for inhibition differing across regions. Our results indicate salience-like signals can have similar downstream consequences to RPE-like signals, although with different temporal dependencies.

*For correspondence:
iwitten@princeton.edu

## Introduction

A critical function of VTA DA neurons is to signal reward prediction error (RPE), or the difference between experienced and expected reward (*Schultz et al., 1997*; *Cohen et al., 2012*). This signal supports reinforcement learning (*Steinberg et al., 2013*; *Chang et al., 2016*; *Tsai et al., 2009*; *Witten et al., 2011*; *Parker et al., 2016*; *Saunders et al., 2018*; *Zweifel et al., 2009*; *Kim et al., 2012*; *Stopper et al., 2014*; *Adamantidis et al., 2011*). However, rather than uniformly representing a scalar RPE, there is recent appreciation that VTA DA neurons can display heterogeneous and spatially organized signals. For example, we recently demonstrated that during navigation-based decision making, there are spatially segregated representations within VTA DA neurons of a variety of sensory, motor, and cognitive variables (*Engelhard et al., 2019*). Others have demonstrated neural correlates of salience-like signals in certain DA neurons (*de Jong et al., 2019*; *Saddoris et al., 2015*; *Wang and Tsien, 2011*; *Gore et al., 2014*; *Yuan et al., 2019*; *Cho et al., 2017*; *Aitken et al., 2016*).

Although most work examining neural correlates of behavior in VTA DA neurons has focused on reward-based tasks, several studies have recorded VTA DA neuron activity during aversive associations (*Robinson et al., 2019*; *Lutas et al., 2019*; *Wang and Tsien, 2011*; *Mileykovskiy and Morales, 2011*). In particular, VTA DA neurons were shown to represent RPE-like signals during fear extinction, in that they display elevated activity when the shock is omitted at the offset of the cue, signaling better-than-expected outcome (*Salinas-Hernández et al., 2018*; *Badrinarayan et al., 2012*; *Jo et al., 2018*). Manipulation of this activity altered the rate of extinction, suggesting that an RPE-like signal in DA neurons drives reinforcement learning for aversive associations as well as rewarding associations (*Salinas-Hernández et al., 2018*; *Luo et al., 2018*).

Here, we first seek to determine if in addition to RPE-like signals, there are also heterogeneous and spatially organized signals in VTA DA neurons during fear extinction. One possibility is that there are neural correlates of salience, and not only RPE, in subregions of the VTA during fear extinction. Salience can be considered an unsigned prediction error - in other words, it is the absolute value of RPE (*Bromberg-Martin et al., 2010*; *Rutledge et al., 2010*). During fear extinction, a neural correlate of salience should have elevated activity during the tone that has been paired with a footshock, and elevated activity at the tone offset, when the expected footshock is not presented.

Spatially segregated RPE versus salience signals in VTA DA neurons would provide a scenario to determine whether or not such distinct signals have similar or different effects on learning, an important question that has not been definitively answered in any behavioral paradigm. One hypothesis is that a salience signal may support learning, similar to an RPE signal. Alternatively, salience signals may acutely modify behavior rather than drive learning.

An advantage of utilizing fear extinction to address these questions is that it provides a continuous readout of learning via a mouse's freezing, allowing us to examine the precise temporal relationship between DA and the expression of learned behavior. However, one limitation of freezing as a readout of learning is that it traditionally requires hand scoring when mice are tethered to neural headgear, as existing software is not designed to dissociate tether movement from mouse movement (*Luyten et al., 2014*; *Shoji et al., 2014*). The need for human labeling has often restricted the analysis of freezing to specific epochs, such as the presentation of the conditioned stimuli.

To overcome this limitation, and provide an automated and unbiased measure of freezing, we developed an analysis pipeline that uses deep learning to automatically identify freezing behavior. We combined this approach with fiber photometry to characterize the spatial distribution of RPE and salience correlates across the medial-lateral axis of the VTA during fear extinction. Finally, we performed optogenetic inhibition of DA neurons in each VTA subregion to assess if, when and how these signals affect fear extinction.

## Results

### Development and application of a convolutional neural network (CNN) to identify freezing behavior

We developed an analysis pipeline based on a convolutional neural network (CNN) to identify freezing behavior in mice. The CNN was initialized on the pre-trained ResNet18 architecture (*He et al., 2016*), and further trained on 'difference images,' the pixel-by-pixel intensity difference between consecutive pairs of frames. The rationale for inputting difference images to the CNN was to capture frame-by-frame motion. Each difference image was human-labeled as 1 or 0 to signify 'freeze' or 'no freeze,' and the network learned to predict labels for new difference images (*Figure 1A–C*, *Figure 1—figure supplement 1A*). We trained a CNN for each of two different experimental chambers (fear conditioning chamber and fear extinction chamber) and two different neural headgears (fiber photometry and optogenetics). Each classifier achieved optimal training within 50 training epochs (*Figure 1—figure supplement 1B–E*) and yielded 92–96% accuracy, 5–10% false positive rate (FPR) and 4–6% false negative rate (FNR) (*Figure 1D*). This was comparable to the relative performance between two humans: given one person's scoring held as ground truth, the other person scored with 95% accuracy, 11.5% FPR and 0.3% FNR (*Figure 1—figure supplement 1F*).

We compared the CNN performance with the popular proprietary software FreezeFrame (*Figure 1E–G*) on an additional 33,000 frames from several mice. Since FreezeFrame produces a second-by-second readout of freezing, we calculated the mean freezing for each second from both the CNN and from human-labeled frames to create a comparable second-by-second readout. We found that the CNN better reflected the human observer than FreezeFrame (*Figure 1E*). The CNN and human observer yielded a correlation of 0.96 (Pearson correlation, *Figure 1F*), while FreezeFrame and human observer yielded a correlation of 0.85 (Pearson correlation, *Figure 1G*). In comparison, the correlation between two human observers was 0.98 (Pearson correlation, *Figure 1—figure supplement 1F*).

Taken together, our pipeline provides an automatic, fast and effective method for scoring freezing. In this paper, we used the CNN to analyze over 500 hrs of behavioral data during fear

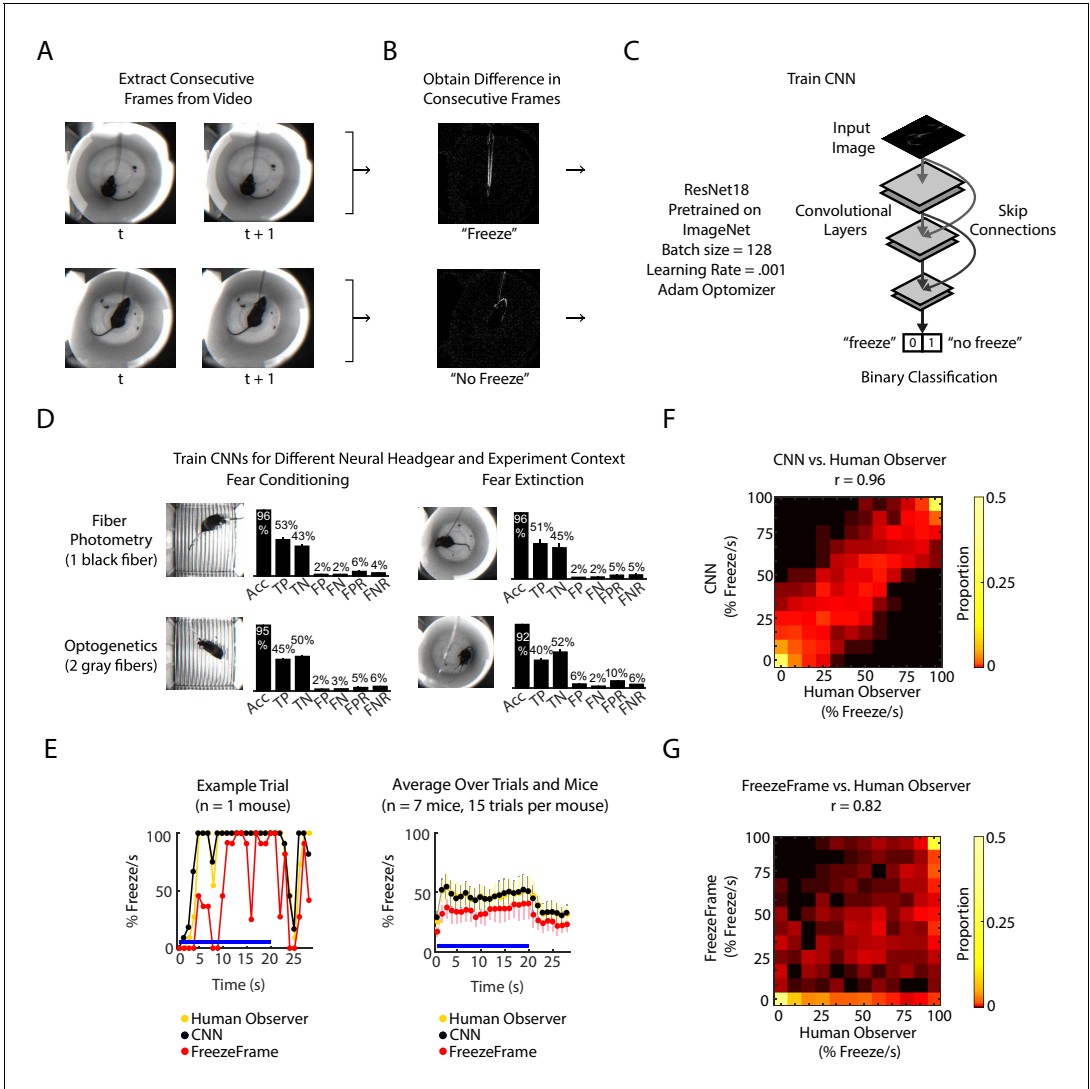

**Figure 1.** CNN classifies frame-by-frame freezing with high accuracy. (**A**) Extract consecutive frames from experiment video. (**B**) Obtain 'difference images,' which are the pixel intensity difference between consecutive frames, and human label a subset as 'freeze' or 'not freeze.' (**C**) Train the CNN classifier to predict 'freeze' or 'no freeze', see Materials and methods for details. (**D**) Train four separate CNNs for two types of neural headgear (fiber photometry and optogenetics) and two experimental backgrounds (fear conditioning and fear extinction). For each CNN, plot shows its accuracy (Acc), false positive rate (FPR), false negative rate (FNR), false negatives (FN), false positives (FP), true negatives (TN), and true positives (TP). Error bars denote SEM of n different training and test datasets (n = 23 for fiber photometry conditioning context, n = 6 for fiber photometry extinction context, n = 15 for optogenetics fear conditioning context, n = 6 for optogenetics extinction context), see Materials and methods and *Figure 1—figure supplement 1* for details. (**E-G**) Comparison between human observer, CNN and FreezeFrame performance on held-out data in the fiber photometry extinction context. (**E**) *Left:* A trial from an example mouse showing percent freezing per second as measured by human observer (yellow), CNN (black), and FreezeFrame (red). For human observer and CNN, percent freezing per second is the mean value of 11 frames where each frame is assigned '1' for 'freeze' and '0' for 'no freeze' relative to previous frame (video acquired at 11.2 Hz). *Right:* Data from all trials and seven mice showing percent freezing per second as measured by human observer (yellow), CNN (black), and FreezeFrame (red). (n = 7 mice, 15 trials per mouse). In both subplots, blue lines denote 20 s tone duration and error bars denote SEM. (**F**) 2-dimensional histogram to compare CNN labeling to human observer (Pearson correlation coefficient r = 0.96, p=0, n = 2940 samples). (**G**) 2-dimensional histogram to compare FreezeFrame labeling to human observer (Pearson correlation coefficient r = 0.82, p=0, n = 2940 samples). (**F-G**) Color intensity denotes proportion of samples in each histogram bin.

The online version of this article includes the following figure supplement(s) for figure 1:

**Figure supplement 1.** CNN performance across different contexts.

conditioning and extinction, which would have been prohibitively time consuming without an automated approach.

## Neural activity in medial and lateral VTA DA neurons during fear extinction correlates with RPE and salience, respectively

We performed fear conditioning and extinction (*Figure 2A*) while simultaneously performing fiber photometry to record from VTA DA neurons. On day 1, mice were presented with ten tones of 20 s duration ('habituation'), followed by ten 20 s tones that coterminated with a 1 s, 0.5 mA foot shock ('conditioning'). On days 2 to 4, mice were presented with twenty-one 20 s tones alone each day ('extinction'). Mice froze very little during habituation, quickly increased freezing during conditioning, and slowly decreased freezing to the tone over three days of extinction (*Figure 2B*).

We sought to determine if in VTA subregions, DA neurons correlated with RPE or salience. During fear extinction, we expect a neural correlate of RPE to have suppressed activity during a tone that has been paired with a foot shock to signal worse than expected outcome, and elevated activity during the tone offset to signal better than expected outcome because the footshock was omitted. A neural correlate of salience, which can be considered an unsigned prediction error, or the absolute value of RPE, should instead have elevated activity during the tone that has been paired with a footshock, and elevated activity at the tone offset, when the shock was unexpectedly omitted (*Figure 2C*).

To perform fiber photometry recordings from VTA DA neurons, we expressed the calcium indicator GCaMP6f in DA neurons by crossing Dat-Cre mice with Ai148D mice (see Materials and methods; *Engelhard et al., 2019*). We targeted the recording fiber to either medial or lateral VTA (*Figure 3A–C*, *Figure 3—figure supplement 1A*; medial VTA group: n = 10 mice, lateral VTA group: n = 11 mice). We chose these subregions because of the electrophysiological, anatomical and functional evidence that there is a medial/lateral distinction within the VTA (*Lammel et al.,*

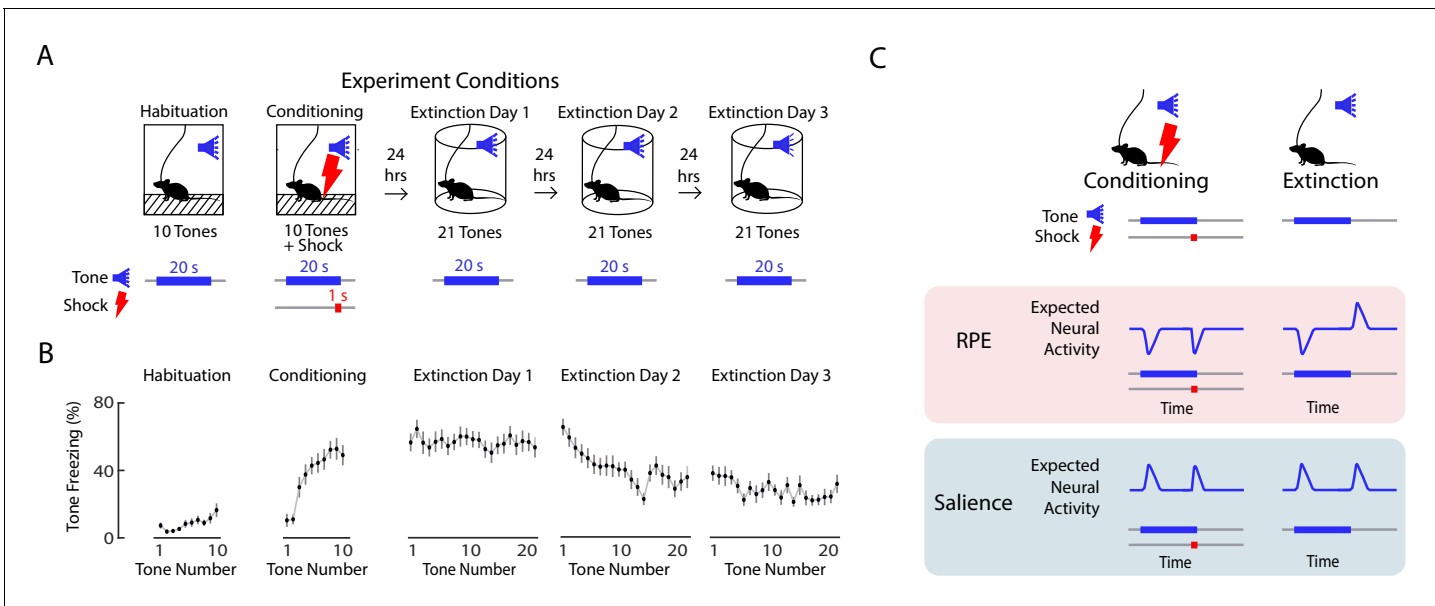

**Figure 2.** Expected neural activity reflecting reward prediction error (RPE) and salience during auditory fear conditioning and extinction. (**A**) Auditory fear conditioning and extinction across 4 days: habituation and fear conditioning occur on the 1 st day, followed by 3 days of extinction. Habituation and conditioning occur in the same experimental chamber, and extinction occurs in a different experimental chamber. During habituation, mice received 10 tones lasting 20 s each. During fear conditioning, mice received 10 tones that coterminated with a 1 s foot shock. During each extinction day, mice received 21 tones. In all conditions, the inter-trial interval was jittered with a mean of 80 s. (**B**) Mean freezing during each tone throughout auditory fear conditioning and extinction (n = 21 mice). Error bars denote SEM. (**C**) Schematic of expected neural activity representing reward prediction error (RPE) and salience during fear conditioning and extinction. During extinction, a neural correlate of RPE would have suppressed activity during a tone that has been paired with a footshock to signal worse than expected outcome, and elevated activity during the tone offset to signal better than expected outcome because the footshock was omitted. A neural correlate of salience can be considered an unsigned prediction error, or the absolute value of RPE.

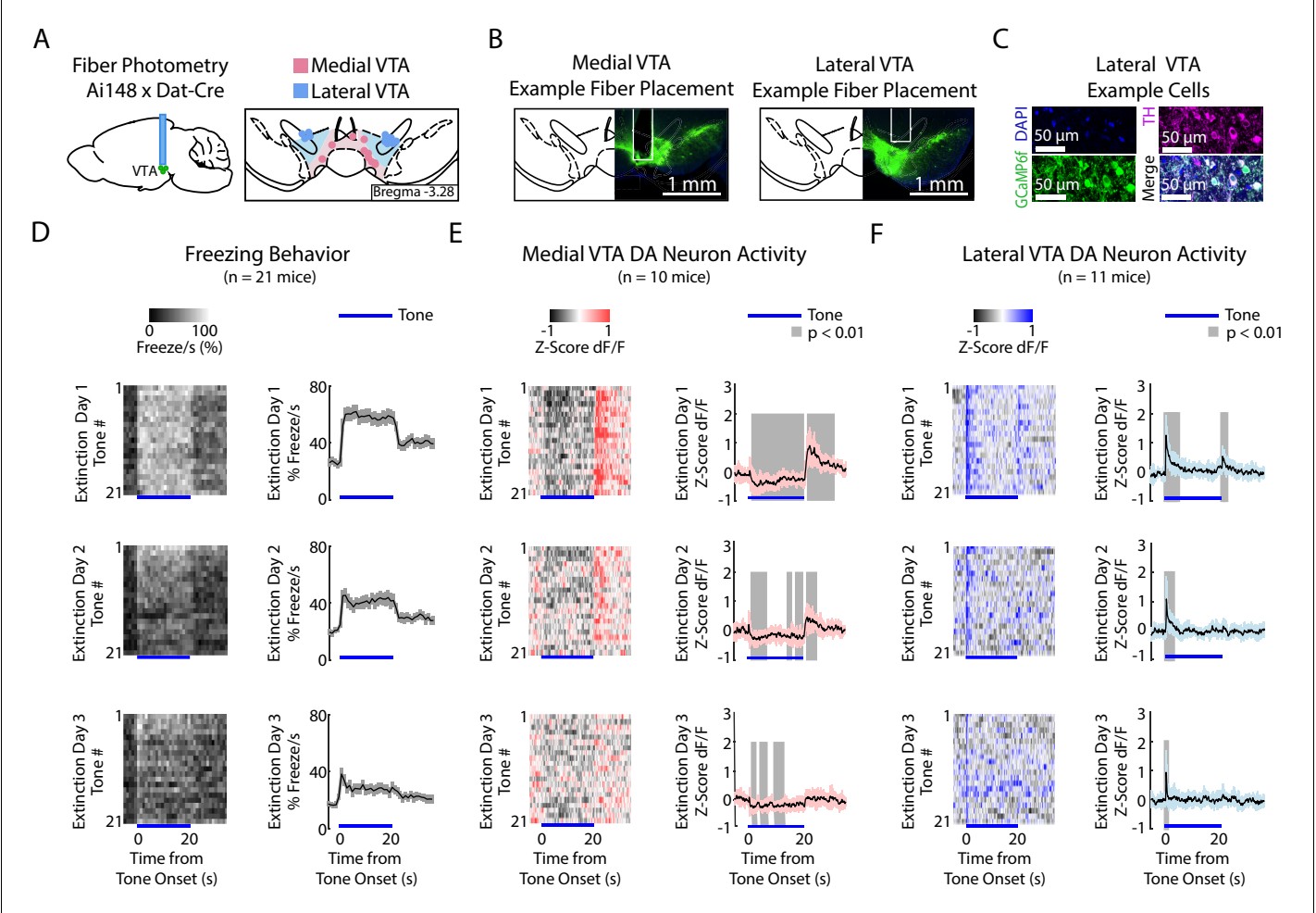

**Figure 3.** During auditory fear extinction, DA neuron activity in medial and lateral VTA resembles RPE and salience, respectively. (**A**) Fiber photometry recordings in medial and lateral VTA. *Left:* Midbrain sagittal slice of fiber implant in VTA. *Right:* Medial and lateral fiber placements across the VTA, visualized at bregma = −3.28 AP. Circles indicate fiber tips, shading indicate approximate boundary of the two regions. (**B**) Example brain slices from different mice showing fiber placement in medial or lateral VTA, at bregma = −3.28 AP. Green denotes GCaMP6f. (**C**) Histology from lateral VTA, staining for cell nuclei (DAPI), tyrosine hydroxylase (TH), and GCaMP6f. (**D-F**) Behavior and VTA DA neuron activity during auditory fear extinction. At the bottom of each subplot, bright blue lines denote tone duration. (**D**) *Left column:* Time course of mean percent freezing for each tone during each day of fear extinction (n = 21 mice). For each extinction day subplot, color intensity corresponds to freezing percentage. *Right column*: Time course of mean percent freezing over all tones for each extinction day (n = 21 mice). Gray shading denotes one standard deviation. (**E**) *Left column:* Time course of mean medial VTA DA neuron activity (GCaMP6f fluorescence) for each tone during each day of fear extinction (n = 10 mice). For each extinction day subplot, color intensity denotes GCaMP6f z-score dF/F. *Right column*: Time course of mean medial VTA DA neuron activity (GCaMP6f fluorescence) over all tones for each extinction day (n = 10 mice). Pink shaded region denotes one standard deviation. Gray shaded region represents 1 s time points where GCaMP6f significantly deviates from shuffled data (percentile rank, p<0.01 after Bonferroni correction for multiple time point comparisons). (**F**) Same as **E** but for lateral VTA DA neuron activity (n = 11 mice), using blue instead of pink shading.

The online version of this article includes the following figure supplement(s) for figure 3:

**Figure supplement 1.** Fiber photometry recordings in medial and lateral VTA dopamine neurons during auditory fear conditioning and extinction.
**Figure supplement 2.** Medial and lateral VTA DA neurons increase activity to both reward (sweetened condensed milk) and foot shock.
**Figure supplement 3.** SNc DA Neurons during fear conditioning and extinction.
**Figure supplement 4.** Medial and lateral VTA DIO-eYFP or DIO-eGFP motion artifacts during auditory fear conditioning and extinction.
**Figure supplement 5.** DA axons in NAcC and DMS during fear conditioning and extinction.

2014; *Beier et al., 2015*; *Yang et al., 2018*; *Engelhard et al., 2019*; *de Jong et al., 2019*). We found that both regions showed increased activity to reward (*Figure 3—figure supplement 2*).

During habituation and fear conditioning, medial and lateral VTA DA neuron signals were similar, but could not be easily explained as purely RPE or salience (*Figure 3—figure supplement 1D,E* top

two rows, and F-H). GCaMP6f fluorescence in both regions decreased throughout the tone during habituation and conditioning. Both regions also showed increased fluorescence to the shock (*Figure 3—figure supplement 1B,C*).

During fear extinction, we observed that medial versus lateral VTA cell bodies preferentially reflected RPE versus salience, respectively (*Figure 3D–F*, *Figure 3—figure supplement 1D,E* bottom three rows). Medial VTA DA neuron activity was consistent with RPE (*Figure 2C*): GCAMP6f fluorescence decreased throughout the tone that had been associated with a negative outcome, and increased at the offset of the tone, during the omission of the expected shock (*Figure 3E*; percentile rank of the mean GCaMP6f fluorescence every second relative to shuffled data: p<0.01 after Bonferroni correction for multiple time point comparison). Both of these signals diminished in magnitude throughout extinction, as expected with RPE.

In contrast, during fear extinction, lateral VTA DA neuron activity was more consistent with salience (*Figure 2C*). GCaMP6f fluorescence increased at the tone onset, consistent with an unsigned prediction error (*Figure 3F*; percentile rank of the mean GCaMP6f fluorescence every second relative to shuffled data: p<0.01 after Bonferroni correction for multiple time point comparisons). GCaMP6f fluorescence also increased at the tone offset, consistent with the idea that the omission of the shock is salient. Both these signals also diminished throughout extinction, as the tone loses its salience.

To control for the possibility that the lateral VTA signal could be coming from the adjacent SNc, in a separate group of mice we recorded from SNc and found different activity profiles (*Figure 3—figure supplement 3*).

To control for possible artifacts in these recordings, we recorded from DAT-Cre mice expressing Cre-dependent GFP (AAV5-DIO-eGFP or AAV5-DIO-eYFP) in medial or lateral VTA. There was little modulation in the signal in the control mice. The exception was the time of the shock, which generated depressed fluorescence, which was the opposite of the increased fluorescence observed with GCaMP6f in medial and lateral VTA (*Figure 3—figure supplement 4*). This suggests that our conclusions above were not due to a recording artifact.

To determine whether dopamine neuron activity in medial and lateral VTA would be similar to dopamine axon activity in the striatum, we recorded from dopamine axons in the nucleus accumbens core (NAcC) and dorsal medial striatum (DMS) using the same fiber photometry technique during the same fear conditioning and extinction behavior. We found that dopamine axons in the nucleus accumbens and DMS both showed RPE during fear extinction, similar to dopamine cell bodies in medial VTA (*Figure 3—figure supplement 5*).

## Medial and lateral VTA DA neuron activity during extinction correlate distinctly with freezing on a trial-by-trial basis and across animals

We next characterized the correlation between freezing and GCaMP6f fluorescence during fear extinction in medial and lateral VTA, both on a trial-by-trial basis, and across animals (*Figure 4*).

In the medial VTA DA neurons, we observed a negative correlation between freezing during the tone and GCaMP6f fluorescence during the tone onset of the same trial (*Figure 4A*; one sample t-test p=0.021, n = 10 mice), but a positive correlation between freezing during the tone and GCaMP6f fluorescence at the tone offset of the same trial (*Figure 4A*; one sample t-test p<$10^{-4}$, n = 10 mice). These correlations are consistent with the interpretation that medial VTA DA encodes an RPE-like signal during fear extinction. Specifically, the negative correlation between fluorescence during the tone and freezing is consistent with the idea that the degree of inhibition in DA during the tone reflects the learned negative association with the tone (*Mileykovskiy and Morales, 2011*). Similarly, the positive correlation between DA after the tone and freezing is consistent with the idea that the fluorescence reflects the degree of 'relief' that the animal experiences when the expected shock is omitted. These same correlations that we observed across trials were also evident when correlating trial-averaged medial VTA GCaMP6f fluorescence and freezing *across animals* (*Figure 4B, C*). Mice with lower average GCaMP6f fluorescence *during* the tone onset or higher average GCaMP6f fluorescence at the tone offset froze more to the tone (Pearson's correlation between GCaMP6f 5 s after tone onset and freezing during tone: r = −0.644, p=0.044; GCaMP6f 5 s after tone offset and freezing during tone: r = 0.768, p=0.010).

In contrast, in the lateral VTA, we observed a *positive* correlation across trials between freezing throughout the tone and GCaMP6f during the tone onset for each trial (*Figure 4D*; one sample

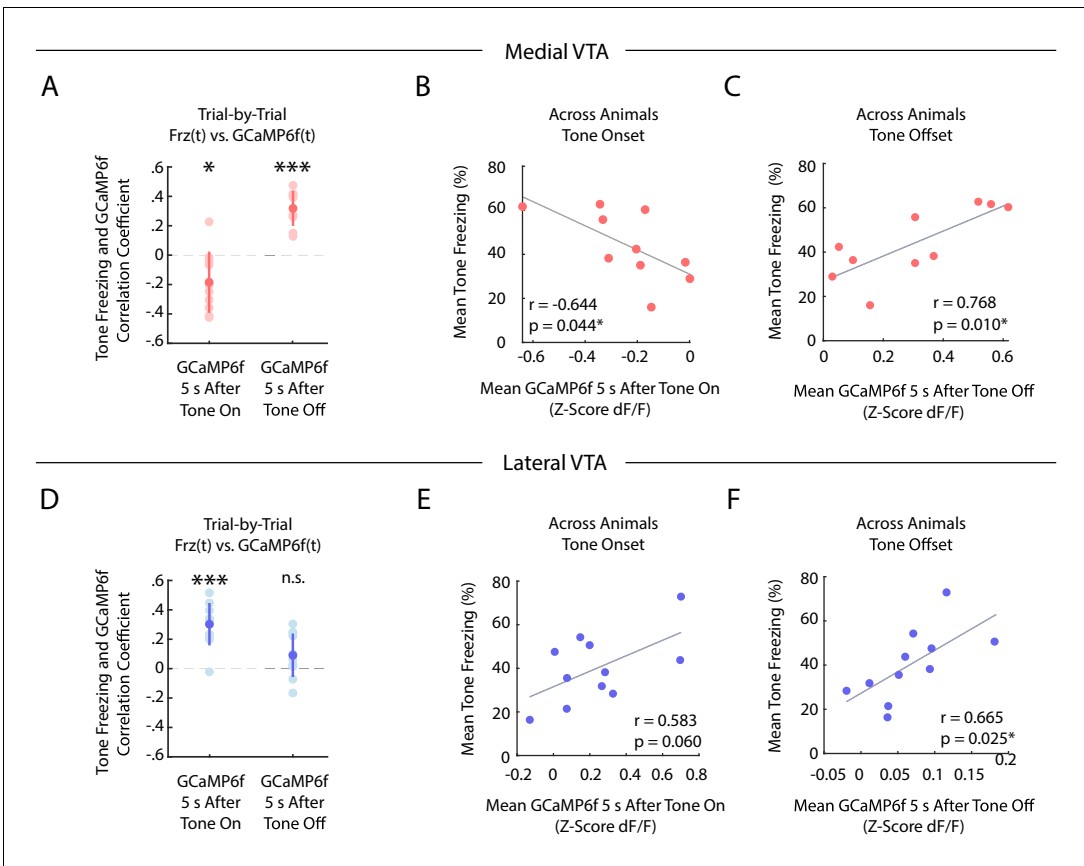

**Figure 4.** During auditory fear extinction, distinct correlations between tone freezing and DA neuron activity in medial and lateral VTA. (**A-C**) Medial VTA trial-by-trial and across animal correlations between freezing and GCaMP6f during fear extinction. In each plot, each dot represents one mouse (n = 10 mice). (**A**) Trial-by-trial correlations: Pearson's correlation coefficient per mouse between mean freezing during each extinction tone and mean GCaMP6f fluorescence 5 s at tone onset (left) or tone offset (right) of the same tone. Error bars denote one standard deviation. Stars denote correlation coefficients that significantly differ from a mean of 0 using one sample t-test (*p<0.05, ***p<0.001). (**B**) Across animal correlations: Correlation across mice between mean freezing during all extinction tones and mean GCaMP6f fluorescence during extinction 5 s at tone onset. Correlation coefficient (r) and p-values on the bottom left of plot; star denotes p<0.05. (**C**) Same as B, but for mean GCaMP6f fluorescence 5 s after tone offset. (**D-F**) Same as (**A-C**), but for lateral VTA trial-by-trial and across animal analysis (n = 11 mice). The online version of this article includes the following figure supplement(s) for figure 4:

**Figure supplement 1.** Lateral VTA DA neuron activity at cue onset correlates with change in freezing, while medial VTA activity does not.

t-test $p<10^{-4}$, n = 11 mice), but not the tone offset (*Figure 4D*; one sample t-test p=0.071). This is consistent with the interpretation that fluorescence during the tone in lateral VTA reflects the salience of the tone, given that more salient stimuli should elicit more freezing. We observed similar trends across mice (*Figure 4E,F*; Pearson's correlation between GCaMP6f during tone onset and freezing during tone: r = 0.583, p=0.060. Pearson's correlation between GCaMP6f after tone offset and freezing during tone: Pearson's correlation, r = 0.665, p=0.025).

In contrast to these medial and lateral VTA trial-by-trial correlations between GCaMP6f and freezing, any correlation between GCaMP6f and change in freezing across trials was much more subtle (*Figure 4—figure supplement 1*).

## At the tone offset, inhibition of medial but not lateral VTA DA neurons slows fear extinction

To investigate whether medial or lateral VTA DA activity contributes to extinction learning, we used optogenetics to inhibit these subregions at specific time points. We injected Cre-dependent NpHR (AAV2/5 DIO-eNpHR3.0-EYFP) or YFP control virus (AAV2/5 DIO-EYFP) into the VTA of DAT-Cre mice and bilaterally implanted optic fibers above medial or lateral VTA.

We first confirmed the efficacy of photoinhibition using whole-cell recordings in acute slices (*Figure 5A-D*). We additionally confirmed photoinhibition in vivo in each subregion by performing real-time place aversion (RTPA) (*Figure 5—figure supplement 1A-C*; medial VTA cohort: NpHR group n = 15, YFP group n = 10. Lateral VTA cohort: NpHR group n = 20, YFP group n = 18), as prior reports show inhibiting VTA DA causes conditioned place aversion (*Calcagnetti and Schechter, 1991*; *Schechter and Meechan, 1994*; *Tan et al., 2012*). Inhibiting either medial or lateral VTA DA caused real time place aversion to the inhibition chamber (*Figure 5—figure supplement 1D*; 2-factor ANOVA with group and subregion as factors: group effect: $F_{(1,1)}$ = 103.33, p = $10^{-14}$; subregion effect: $F_{(1,1)}$ = 0.32, p = 0.576; group × subregion interaction: $F_{(1,1)}$ = 0.01, p = 0.935) and decreased velocity during inhibition (*Figure 5—figure supplement 1E*; 2-factor ANOVA with group and subregion as factors: group effect: $F_{(1,1)}$ = 14.35, p < $10^{-3}$; subregion effect: $F_{(1,1)}$ = 0.1, p = 0.754; group × subregion interaction: $F_{(1,1)}$ = 0, p = 0.9441).

We next examined the effect of VTA DA neuron inhibition during fear extinction in separate cohorts of mice with fibers in medial or lateral VTA; each cohort had separate groups of NpHR- or YFP-virus injected mice (*Figure 5E–K*, *Figure 5—figure supplement 2A–B*; medial VTA cohort: NpHR group n = 14, YFP group n = 10. Lateral VTA cohort: NpHR group n = 8, YFP group n = 8). Both cohorts underwent fear conditioning, and during fear extinction, received inhibition lasting for 6 s starting from the last second of the tone to 5 s after the tone off.

Inhibiting medial VTA DA neurons at tone offset yielded an effect consistent with the representation of RPE in this subregion: NpHR mice froze more to the tone compared to the YFP controls (*Figure 5F-H*; 2-factor ANOVA with group and tone number as factors, tone number being a repeated measure: group effect: $F_{(1,62)}$ = 7.528, p = 0.012; group × tone number interaction: $F_{(1,62)}$ = 1.056 p = 0.361). This was consistent with previous work that showed inhibiting medial VTA during cue offset impairs extinction learning (*Luo et al., 2018*; *Salinas-Hernández et al., 2018*). In addition to attenuating extinction of tone-induced freezing, NpHR mice increased freezing during and after the inhibition period compared to YFP controls (2-factor ANOVA with group and tone number as factors, tone number being a repeated measure. During 6 s inhibition period: group effect: $F_{(1,62)}$ = 17.668, p < $10^{-3}$; group × tone number interaction: $F_{(1,62)}$ = 1.371 p = 0.032. During 6 s after inhibition period: group effect: $F_{(1,62)}$ = 28.082, p < $10^{-4}$; group × tone number interaction: $F_{(1,62)}$ = 0.854 p = 0.782.) Thus, not only do medial VTA DA neurons lead to updating of the value (or freezing behavior) upon subsequent presentations of the tone that preceded inhibition, they also modify behavior subsequent to the inhibition period.

In contrast to the effects observed with medial VTA inhibition, inhibiting the lateral VTA DA neurons at tone offset yielded no change in freezing during the tone (*Figure 5I-K*; 2-factor ANOVA with group and tone number as factors, tone number being a repeated measure: group effect: $F_{(1,62)}$ = 0.002, p = 0.964; group × tone number interaction: $F_{(1,62)}$ = 1.121 p = 0.249). This manipulation did, however, increase freezing after the inhibition period, an effect that increased with extinction (2-factor ANOVA with group and tone number as factors, tone number being a repeated measure: group effect: $F_{(1,62)}$ = 6.981, p = 0.019; group × tone number interaction: $F_{(1,62)}$ = 1.480 p = 0.011). This suggests that lateral VTA inhibition primarily affects freezing at time points after inhibition, rather than causing learned changes in the value of preceding events.

We performed analogous optogenetic experiments in mice with fibers targeting the SNc (*Figure 5—figure supplement 3A-C* and *Figure 5—figure supplement 2D*, NpHR group n = 7, YFP group n = 6). Similar to the lateral VTA cohort, inhibiting SNc DA neurons at tone offset did not have an effect on tone freezing. However, SNc NpHR mice showed decreased freezing during the ITI relative to the YFP group (*Figure 5—figure supplement 3D*; 2-factor ANOVA with group and tone number as factors, tone number being a repeated measure: group effect: $F_{(1,62)}$ = 4.980, p = 0.047; group × tone interaction: $F_{(1,62)}$ = 1.276, p 0.081), which we did not observe with the lateral

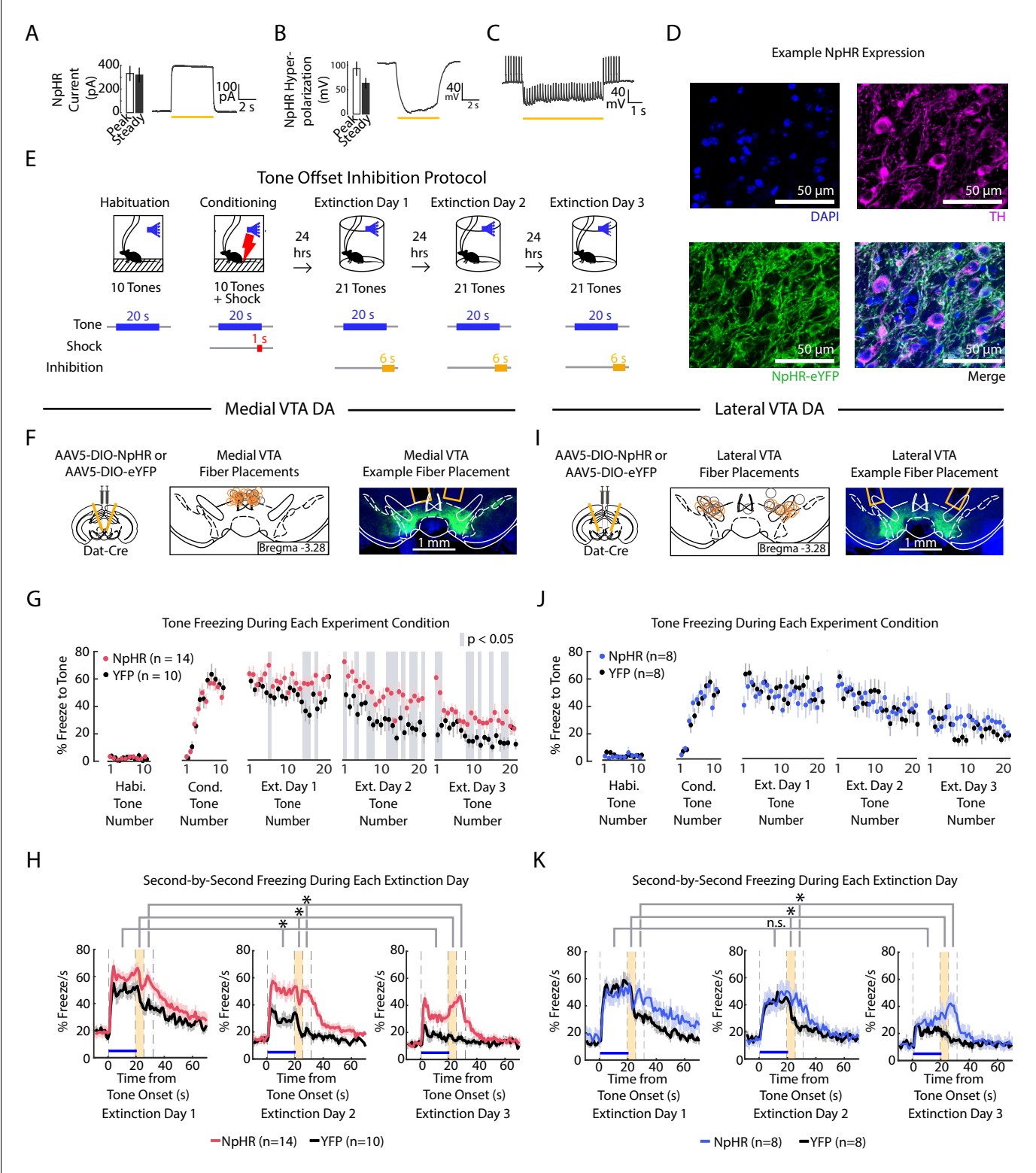

**Figure 5.** Optogenetic inhibition of medial but not lateral VTA DA neurons at tone offset slows auditory fear extinction. (**A-C**) Photoinhibition of NpHR-expressing VTA DA neurons during ex vivo whole-cell recordings. (**A**) *Left:* Mean and SEM of photocurrents evoked during voltage clamp (n = 7 neurons; peak current: 332.15 ± 64.79 pA; steady state current: 318.84 ± 63.64 pA). *Right:* Example trace of photocurrent. (**B**) Mean and SEM of NpHR-mediated hyperpolarization during current clamp (n = 7 neurons; peak hyperpolarization from baseline: 92.76 ± 14.58 mV; steady state hyperpolarization from baseline: 62.93 ± 10.92 mV). Baseline resting potentials ranged from -50 to -60 mV. *Right:* Example trace of NpHR-mediated

*Figure 5 continued on next page*

*Figure 5 continued*

hyperpolarization. Baseline potential is -60 mV. (**C**) Example trace of photoinhibition of spikes generated by current injections (200 pA injections) during current clamp recording. Baseline potential is -57 mV. Yellow bar denotes photostimulation period. (**D**) Histology of example lateral VTA neurons showing cell nuclei (DAPI), tyrosine hydroxylase (TH), NpHR-eYFP. (**E**) Optogenetic inhibition protocol. Laser inhibition (594 nm) occurs for 6 s for every extinction tone, starting 1 s prior to tone offset and ending 5 s after tone offset. (**F-H**) Medial VTA DA neuron tone offset inhibition during fear extinction (n = 14 NpHR mice, n = 10 YFP mice). (**F**) *Left:* Schematic of injection of an AAV2/5 virus expressing Cre-dependent eNpHR3.0-YFP (or YFP-only virus) into the VTA of Dat-Cre mice, with fiber implant into medial VTA. *Middle:* medial VTA fiber placements, visualized at bregma = -3.28 AP. Circles indicate fiber tips, yellow circles indicate NpHR mice, grey circles indicate YFP mice. For full AP coordinates of fiber placements, see *Figure 5—figure supplement 2A*. *Right:* Example brain slice showing fiber placements in medial VTA. Green denotes NpHR3.0-eYFP. (**G**) Freezing during the tone for each tone number, by experiment condition: habituation, conditioning, extinction day 1, extinction day 2 and extinction day 3. Pink dots denote NpHR group, and black dots denote YFP control group. Error bars denote SEM. Gray shading denotes tone trials that significantly differ between NpHR and YFP group (post hoc student's unpaired t-test, p < 0.05). (**H**) Percent freezing per second averaged across all tones for each extinction day during medial VTA tone offset inhibition (pink line denotes NpHR group, black line denotes YFP control group). Yellow shaded region denotes the inhibition period. Error bars denote SEM. Stars denote a significant group or interaction effect using a 2-factor ANOVA with group and tone number as factors, tone number being a repeated measure. This analysis was applied to predict freezing during the tone (group effect: $F_{(1,62)}$ = 7.528, p = 0.012; group × tone number interaction: $F_{(1,62)}$ = 1.056 p = 0.361), freezing during the 6 s inhibition period (group effect: $F_{(1,62)}$ = 17.668, p = $10^{-3}$; group × tone number interaction: $F_{(1,62)}$ = 1.371 p = 0.032), and freezing 6 s following the inhibition period (group effect: $F_{(1,62)}$ = 28.082, p < $10^{-4}$; group × tone number interaction: $F_{(1,62)}$ = 0.854 p = 0.782). (**I**) Same as **F** except for lateral VTA (n = 8 NpHR mice, n = 8 YFP mice). For full AP coordinates of fiber placements, see *Figure 5—figure supplement 2B*. (**J-K**) Same as (**G-H**), but for lateral VTA DA neuron tone offset inhibition during fear extinction, with blue representing the NpHR group and black representing YFP group. (**J**) Post hoc analysis was not conducted because the 2-factor ANOVA in **K** did not yield a significant group effect on tone freezing. (**K**) Stars denote a significant group or interaction effect using a 2-factor ANOVA with group and tone number as factors, tone number being a repeated measure. This analysis was applied to predict freezing during the tone (group effect: $F_{(1,62)}$ = 0.002, p = 0.964; group × tone number interaction: $F_{(1,62)}$ = 1.121 p = 0.249), freezing during the 6 s inhibition period (group effect: $F_{(1,62)}$ = 1.150, p = 0.302; group × tone number interaction: $F_{(1,62)}$ = 1.387 p = 0.029), and freezing 6 s following the inhibition period (group effect: $F_{(1,62)}$ = 6.981, p = 0.019; group × tone number interaction: $F_{(1,62)}$ = 1.480 p = 0.011).
The online version of this article includes the following figure supplement(s) for figure 5:

**Figure supplement 1.** Medial or lateral VTA DA neuron inhibition causes real time place aversion and decreased velocity.
**Figure supplement 2.** Medial VTA, lateral VTA, and SNc fiber placements for optogenetic inhibition.
**Figure supplement 3.** Optogenetic inhibition of SNc DA neurons at tone offset.

VTA cohort. This suggests that lateral VTA and SNc inhibition during the cue offset may not have identical consequences.

## Inhibition of lateral VTA at the tone onset slows extinction learning

We next inhibited lateral VTA DA neurons during fear extinction at the tone onset, given that is when the fiber photometry recordings showed a pronounced salience-like signal in that subregion (*Figure 6A, B*, *Figure 5—figure supplement 2C*; NpHR group n = 12, YFP group n = 10; 6 s inhibition period). We found that NpHR mice extinguished more slowly to the tone. This is reflected in that there was no main effect of group, but a significant interaction between group and tone number (*Figure 6C, D*; 2-factor ANOVA with group and tone number as factors, tone number being a repeated measure: group effect: $F_{(1,62)}$ = 0.014, p = 0.908; group × tone number interaction: $F_{(1,62)}$ = 2.217, p < $10^{-6}$). Additionally, we observed that NpHR mice showed decreased freezing during the ITI (*Figure 6D*; 2-factor ANOVA with group and tone number as factors, tone number being a repeated measure: group effect: $F_{(1,62)}$ = 6.477, p = 0.019; group × tone number interaction: $F_{(1,62)}$ = 1.562 p = 0.004).

Together with the results from the previous experiments, this suggests that inhibiting the salience-like signal at tone onset slows extinction learning over time, even though activity of these neurons at tone offset does not update the value of the *preceding* tone.

## Discussion

DA neurons originating in the VTA are known to be modulated by aversive stimuli, and have been implicated in fear conditioning and extinction (*El-Ghundi et al., 2001*; *Young et al., 1993*; *Inoue et al., 2000*; *Nader and LeDoux, 1999*; *Guarraci and Kapp, 1999*; *Holtzman-Assif et al., 2010*; *Delgado et al., 2008*; *Luo et al., 2018*; *Salinas-Hernández et al., 2018*; *Zweifel et al., 2011*; *Pezze and Feldon, 2004*; *Mueller et al., 2010*; *Pignatelli et al., 2017*; *Nasehi et al., 2016*;

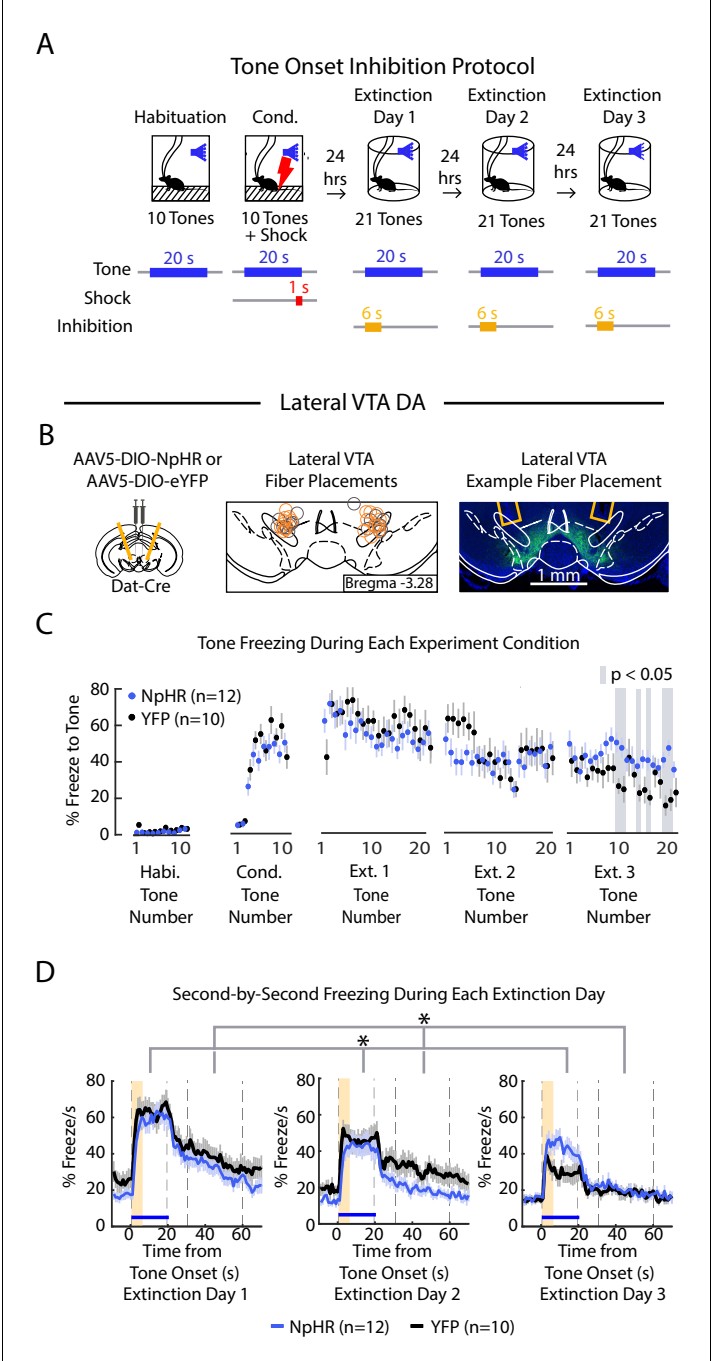

**Figure 6.** Optogenetic inhibition of lateral VTA DA neurons at tone onset slows auditory fear extinction.
(**A**) Optogenetic inhibition protocol. Laser (594 nm) inhibition occurs simultaneously with tone onset for 6 s. (**B**)
Lateral VTA DA neuron tone onset inhibition during fear extinction (n = 12 NpHR mice, n = 10 YFP mice). *Left:*
Schematic of injection of an AAV2/5 virus expressing Cre-dependent eNpHR3.0-YFP (or YFP-only virus) into the
VTA of Dat-Cre mice, with fiber implant into lateral VTA. *Middle:* lateral VTA fiber placements, flattened at bregma
= -3.28 AP. Circles indicate fiber tips, yellow circles indicate NpHR mice, grey circles indicate YFP mice. For full AP
coordinates of fiber placements, see *Figure 5—figure supplement 2C*. *Right:* Example brain slice showing fiber
placements in lateral VTA. Green denotes NpHR3.0-eYFP. (**C**) Freezing during the tone for each tone number, by
experiment condition: habituation, conditioning, extinction day 1, extinction day 2 and extinction day 3. Blue dots
denote NpHR group, and black dots denote YFP control group. Error bars denote SEM. Gray shading denotes
tone trials that significantly differ between NpHR and YFP group (post hoc student's unpaired t-test, p < 0.05). (**D**)
Percent freezing per second averaged across all tone trials for each extinction day during lateral VTA tone onset

*Figure 6 continued on next page*

*Figure 6 continued*

inhibition (blue line denotes NpHR group, black line denotes YFP control group). Yellow shaded region denotes inhibition period. Error bars denote SEM. Stars denote a significant interaction effect using a 2-factor ANOVA with group and tone number as factors, tone number being a repeated measure. This analysis was applied to predict freezing during the tone: group effect: $F_{(1,62)} = 0.014$, $p = 0.908$; group $\times$ tone interaction: $F_{(1,62)} = 2.217$, $p < 10^{-6}$, and freezing 30 s during the ITI: group effect: $F_{(1,62)} = 6.477$, $p = 0.019$; group $\times$ tone number interaction: $F_{(1,62)} = 1.562$ $p = 0.004$.

---

*Pezze et al., 2003*; *Budygin et al., 2012*; *Robinson et al., 2019*; *Lammel et al., 2011*; *Jo et al., 2018*; *Lutas et al., 2019*; *Fadok et al., 2009*; *Groessl et al., 2018*; *Bouchet et al., 2018*; *Wenzel et al., 2018*; *Wang and Tsien, 2011*; *Mileykovskiy and Morales, 2011*). However, it was unclear if and how neural correlates of fear extinction are topographically organized within the VTA. In addition, the causal contribution of spatially localized DA activity within the VTA to fear extinction was unknown. Here, we found that during fear extinction, medial VTA more closely resembled RPE, while lateral VTA more closely resembled salience. While activity in both subregions contributed causally to fear extinction, the temporal relationship between activity and freezing differed. Consistent with the idea that RPE signals update the value of preceding events, inhibition in medial VTA prevented the update of value (or freezing response) to the tone preceding the inhibition, causing freezing to be more likely on subsequent tone presentations. In contrast, inhibition of the salience-like signal in lateral VTA affected freezing in the time period during or immediately following the inhibition, but did not update the value (or freezing response) to subsequent presentations of the tone if it preceded the inhibition.

## Development and application of a CNN to identify freezing behavior

Learned fear is typically quantified by measuring a mouse's freezing. When mice are tethered to neural headgear, existing automated algorithms cannot accurately dissociate movement of the mouse from movement of the tether. To solve this problem, we turned to deep learning, and used an off-the-shelf CNN architecture, ResNet, known for its applications in image classification and pose estimation (*Nath et al., 2019*; *Insafutdinov et al., 2016*; *He et al., 2016*; *Mathis et al., 2018*). Our CNN allowed accurate and automated classification of freezing behavior throughout the duration of our experiments with minimal labor, and enabled us to determine that the precise temporal relationship between dopamine neuron activity and freezing behavior depended on VTA subregion.

## Spatial organization of RPE and salience-like signals in the medial versus lateral VTA during fear extinction

Our finding of RPE-like signals in medial VTA and salience-like signals in lateral VTA during fear extinction may reflect a larger organizational structure across both the VTA and the substantia nigra pars compacta (SNc) dopamine neurons. Specifically, there is previous evidence that the SNc, which is lateral to the VTA, has a greater proportion of DA neurons that signal salience rather than RPE (*Matsumoto and Hikosaka, 2009*; *Menegas et al., 2017*). One question is how the presence of salience signals in lateral VTA and SNc may relate to other work showing correlates of kinematics in addition to RPE in those same regions (*Engelhard et al., 2019*; *da Silva et al., 2018*; *Howe and Dombeck, 2016*). One possible relationship between these seemingly disconnected findings of salience versus kinematic tuning is that an animal's speed is a reflection of the motivational salience of an environment (*Zenon et al., 2016*; *Panigrahi et al., 2015*; *Wang et al., 2013*).

It is worth noting that this medial/lateral distinction between RPE and salience does not hold across all behaviors (*Heymann et al., 2020*). In fact, we found that during fear conditioning, medial and lateral VTA shared similar activity profiles, which may reflect heterogeneity in single neuron activity that is not captured with fiber photometry. Furthermore, it is not obvious how to integrate our spatial findings with recent DA terminal recordings within the NAc, despite the rough topography between VTA cell bodies and their projections (*Yang et al., 2018*; *de Jong et al., 2019*; *Beier et al., 2015*). In particular, recent studies reported increased activity to aversive stimuli in ventromedial NAc, and decreased activity to aversive stimuli in lateral NAc (*de Jong et al., 2019*; *Yuan et al., 2019*). Assuming topography between VTA cell bodies and their terminals in NAc, we may have expected the opposite result. This discrepancy could be due to a number of factors,

including imperfect topography between cell bodies and terminals, differences in the behavioral paradigm, or terminal activity that does not reflect cell body activity (*Berke, 2018*; *Threlfell et al., 2012*).

### Distinct temporal relationships between activity in the medial and lateral VTA and fear extinction

Our manipulations of medial VTA DA neurons produced changes in extinction learning that were consistent with the presence of an RPE-like signal in that subregion. Specifically, the burst of DA at the omission of the expected shock contributed causally to extinction learning, a finding which aligned with previous work (*Salinas-Hernández et al., 2018*; *Luo et al., 2018*).

In addition, this manipulation increased freezing in the time period immediately after the inhibition, suggesting that not only do these neurons regulate learning about the value of a preceding event (the tone), they also affect freezing behavior in the time period subsequent to the manipulation.

While our results in medial VTA were consistent with evidence that RPE signals in DA neurons support reinforcement learning in a variety of paradigms (*Steinberg et al., 2013*; *Witten et al., 2011*; *Chang et al., 2016*; *Zweifel et al., 2009*; *Tsai et al., 2009*; *Kim et al., 2012*; *Stopper et al., 2014*; *Adamantidis et al., 2011*), the causal contribution of salience signals in DA neurons to behavior is much less clear. By leveraging the spatial organization we uncovered between RPE versus salience signals in medial versus lateral VTA, we could directly examine the causal role of salience-like signals. One possibility is that, similar to RPE, salience signals in lateral VTA may also support learning (*Bromberg-Martin et al., 2010*).

We found that inhibition of the lateral VTA at the tone offset did not influence extinction learning of the preceding tone. This was consistent with the previous observation that inhibition of a subpopulation of VTA neurons with projections to NAc Core did not affect extinction (*Luo et al., 2018*). However, similar to medial VTA, this manipulation increased freezing in the timepoints directly after the inhibition, suggesting that despite the fact that these neurons did not regulate learning about the value of a preceding event (the tone), they did affect behavior in the time period subsequent to the manipulation. Consistent with this manipulation affecting subsequent timepoints, inhibition at the tone onset, when these neurons have a burst of activity, slowed the rate of extinction. However, we cannot rule out that this manipulation affected the expression of freezing rather than extinction learning. Additionally, this lateral VTA tone onset inhibition had more subtle effects on behavior relative tomedial VTA tone offset inhibition. .

In summary, during fear extinction, medial and lateral VTA DA neurons provide distinct but complementary signals that contribute to extinction learning, but at different times. Medial VTA encodes an RPE-like signal, which serves to update the value of the preceding tone. Lateral VTA encodes a salience-like signal at the tone onset, which does not update the value of the preceding tone, but affects freezing during the tone. This is consistent with the emerging framework that despite differences in neural correlates, different VTA/SNc DA subpopulations share a common function of mediating learning, even though they may differ in the specific setting in which they contribute to learning, or the specific aspect of learning that they mediate (*Ellwood et al., 2017*; *Saunders et al., 2018*; *Cox and Witten, 2019*; *Bromberg-Martin et al., 2010*; *Menegas et al., 2018*).

## Materials and methods

### Animals

All experiments followed guidelines established by the National Institutes of Health and reviewed by Princeton University Institutional Animals Care and Use Committee (IACUC). Dat-Cre mice with IRES-Cre inserted just 3' of the termination colon of the *Slc6a3* gene that encodes the dopamine transporter (B6.SJL-Slc6a3tm1.1(cre)Bkmn/J, Jackson Labs) were bred in house with Ai148D mice (Ai148(TIT2L-GC6f-ICL-tTA2), Jackson Labs), and male progeny expressing GCaMP6f in dopamine neurons (Dat-Cre x Ai148D mice) were used for fiber photometry experiments. Male Dat-Cre mice were used for optogenetic experiments and electrophysiology recordings. Mice were group housed with up to five other mice, allowed ad libitum access to food and water, and kept on a 12 hr light on

and 12 hr light-off schedule. Mice between 7–10 weeks of age were used during surgery. We conducted all surgery and behavioral experiments during light off period.

## Stereotactic surgeries

Mice were induced to a surgical plane of anaesthesia using 4% isoflurane, and maintained at. 0.5–2% isoflurane for the duration of the surgery. Mice were kept warm using a heating pad, and breathing rate was monitored by the surgeon. At surgery onset, mice received i.p. injections of nonsteroidal anti-inflammatory drug meloxicam (2 mg/kg) and the antibiotic baytril (5 mg/kg). Twenty-four hours post surgery, mice received a second equivalent dose of meloxicam.

For fiber photometry experiments, we used transgenic Dat-Cre x Ai148D male mice which expressed GCaMP6f in dopamine neurons. Optic fibers of 400 µm core diameter (Mono Fiberoptic Cannula, Doric Lenses) were implanted unilaterally either in the medial VTA (-3.1 mm posterior, +/-0.5 mm lateral, -4.5 mm ventral, n=10 mice), lateral VTA (-3.1 mm posterior, +/- 0.75 mm lateral, -4.2 mm ventral, n=11 mice), or SNc (-2.9 posterior, +/-1.25 lateral, -4.2 ventral relative to Bregma n = 2 mice), NAcC (+1.2 anterior, +/-1.0 lateral, -4.5 ventral relative to Bregma, n = 7 mice) or DMS (+0.74 anterior, +/-1.4 lateral, -2.6 ventral relative to Bregma, n = 13 mice). Fibers were affixed to the skull using dental cement (C&B-Metabond). To control for motion artifacts in fiber photometry experiments, we used a nanosyringe (World Precision Instruments) to inject Dat-Cre mice with 600 nL of virus containing Cre-dependent green or yellow fluorescent protein (AAV2/5-CAG-FLEX-eGFP-WPRE-bGH (Allen Institute) or AAV2/5-EF1a-DIO-eYFP-hGHpA (UPenn or Princeton Vector Core)) into the VTA (-3.1 mm posterior, +/-. 5 mm lateral, -4.75 mm ventral). Virus was injected bilaterally and needle remained in position post injection for 6 min to allow injection to disperse before removal. We implanted optic fibers in the medial or lateral VTA as mentioned above (medial VTA n = 4 mice, lateral VTA n = 4 mice).

For optogenetic experiments, we used a nanosyringe (World Precision Instruments) to deliver 600 nL of virus containing Cre-dependent halorhodopsin AAV2/5-EF1a-DIO-eNpHR3.0-EYFP-WPRE-hGHpA (UPenn or Princeton vector core, $10^{13}$ to $10^{14}$ parts/ml) or the control virus AAV2/5-EF1a-DIO-eYFP-WPRE-hGHpA (Upenn and Princeton vector core, $10^{13}$ to $10^{14}$ parts/ml) into the brain. Virus was injected bilaterally into the medial VTA (−3.1 mm posterior, +/- 0.5 mm lateral, −4.75 mm ventral, n = 26 mice, needle bevel pointed posterior), lateral VTA (−3.1 mm posterior, +/- 0.75 mm lateral, −4.75 mm ventral, n = 28 mice, needle bevel pointed posterior), or SNc (−2.9 mm posterior, +/- 0.75 mm lateral, −4.70 mm ventral, n = 13 mice, needle bevel pointed lateral); each injection was 600 nL in volume and injected at 100 nL/min, and the needle remained in position for 6 min post injection to allow injection to disperse before removal. Mice were then bilaterally implanted respectively with optic fibers (300 um core diameter) in the medial VTA (−3.1 mm posterior, +/-0.1.09 mm lateral, −4 mm ventral at 10 degree angle; no-angle exact coordinates −3.1 mm posterior, +/- 0.5 mm lateral, −4.5 ventral), lateral VTA (−3.1 mm posterior, +/- 0.1.69 mm lateral, −4 mm ventral at 10 degree angle; no-angle exact coordinates: −3.1 mm posterior, +/- 0.75 mm lateral, −4.2 mm ventral), or SNc (−2.9 mm posterior, −1.9 mm lateral, −3.8 mm ventral at 10 degree angle; no angle coordinates: −2.9 mm posterior, −1.25 mm lateral, −4.0 mm ventral relative to Bregma). Optic fibers were affixed to the skull using dental cement.

## Behavioral assays

### Auditory fear conditioning and extinction

Following a minimum of 6 days to recover from cranial surgery, mice 8–12 weeks of age underwent auditory fear conditioning and extinction. We performed this assay using FreezeFrame (Coulbourn Instruments), and throughout the assay gray-scaled video was recorded at 11.2 Hz while mice were tethered to either fiber photometry or optogenetic cables for neural recording or manipulation, respectively. On day one, mice were placed in a chamber (10 cm x 10 cm) with a conductive metal floor and a smell of ethanol. Mice were habituated to ten 20 s tones (5 kHz, 70 dB), followed by ten 20 s tones that coterminated with a 1 s foot shock administered through the metal floor (1 mA, scrambled). On days two to four, mice underwent extinction in a different but similarly sized experimental chamber with plastic floor and walls, and a different smell (quatricide). During extinction, mice heard 21 tones per day without the shock. For all experiment days, the inter-trial interval (ITI)

between tones was jittered with a Poisson distribution with a mean of 80 s; the ITI was randomly selected prior to all experiments and was the same for each mouse.

## Real-time place preference (RTPP)

Prior to, or following the fear conditioning and extinction assay, mice underwent real-time place preference assay, counter-balanced between groups. Mice were connected to optical fibers and individually placed in a rectangular enclosure with two chambers (28 cm x 28 cm each), and allowed to freely explore the two chambers for 20 min. Gray-scaled video was recorded at 30 fps and mouse location and velocity was tracked throughout experiment using Ethovision XT 9. For the first 5 min, the mice freely explored both chambers but they did not receive optical stimulation ('baseline'). For the next 15 min, mice continued to have access to both chambers, but the presence in one chamber resulted in continuous delivery of laser light (594 nm, 6 mW, Cobalt) to the implanted optic fibers. The 'light on' chamber was randomly assigned and counterbalanced within each experimental group (NpHR or YFP).

## Random reward delivery

Mice with regular ad libitum access to food and water were put in a 21 cm ×18 cm modular operant chamber (MED Associates, ENV-307W). 12 uL of sweetened condensed milk (Carnation, diluted 1:10) were delivered 5-10 times at random intervals ranging from 60 to 200 s. Mouse licking was detected if the mouse's body closed the loop on an open circuit, where one end of the circuit was the metal floor of the operant chamber, and the other end of the circuit was the reward delivery spout; the licking signal was acquired at 25 Hz. Fiber photometry fluorescence signal was also acquired at 25 Hz (instead of 100 Hz in all other fiber photometry recordings).

## Fiber photometry experiments

One to two weeks after surgery, mice individually underwent fear conditioning and extinction during fiber photometry recording of VTA DA neurons expressing GCaMP6f. Mice were connected to a fiber photometry set up described in previous reports (*Gunaydin et al., 2014*). A 488 nm laser light (Micron Technology) was filtered (FL488, Thor Labs) then passed through a dichroic mirror (MD498, Thor Labs) and traveled through a patch cable (Mono Fiberoptic Patchcord, 400 um core, 0.48 Numerical Aperture, Doric Lenses) coupled via a ceramic split sleeve (2.5 mm diameter, Precision Fiber Products) to the optic fiber implanted in the mouse brain; the light traveled down the optic fiber into the VTA for fluorescence excitation. Laser light delivery was controlled by a lock-in amplifier (Ametek, 7265 Dual Phase DSP Lock-in Amplifier), which delivered light at 210.999 Hz, and the laser intensity at the tip of the patch cable was approximately 5 uW. Fluorescent emission from GCaMP6f at 500–550 nm was then passed through the same patch cable, filtered (MF525-39, Thor Labs), and passed through the same dichroic mirror into a photodetector (Model 2151, New Focus), and the signal was filtered at the same 210.999 Hz using the same lock-in amplifier, and a time constant of 20 ms. AC gain on the lock-in amplifier was set to 0 dB. Signal was digitized at 100 Hz. dF/F was calculated by the following formula:

$$\frac{dF}{F} = \frac{F - F_0}{F_0}$$

Where $F_0$ is the second order polynomial fit to the GCaMP6f signal for the duration of the experimental session, in order to account for very slow decline of the signal over time, presumably caused by photobleaching. Z-score of dF/F signal was calculated across experiment days for each mouse; the mean used for Z-score is the mean signal across all 4 experiment days, and standard deviation used for Z-score is the standard deviation across all 4 days. To obtain shuffled data, GCaMP6f fluorescence from each session was circularly shifted across the session in time by a random time bin (n = 1000 shuffles unless otherwise specified). The Bonferroni corrections used in *Figure 3E, F*, *Figure 3—figure supplement 1G, H, Figure 3—figure supplement 3C; Figure 3—figure supplement 4D,E*, were corrected by 40 time points to account for the 40 1-second-long bins. The Bonferroni corrections used in *Figure 3—figure supplement 2* were corrected with 10 time points to account for the 10 1-second-long bins.

One lateral VTA fiber photometry mouse was excluded from analysis because the fiber location was at the SNc border.

## Optogenetic experiments

Approximately 1 month after opsin or control virus injection, mice underwent fear conditioning and extinction. Littermates within a cage were randomly allocated to either NpHR or YFP group, with as equal distribution as possible. (Ie. in a cage with four mice: two mice were in NpHR group, two mice in YFP group. In a cage with five mice, two mice were in NpHR group and three mice in YFP group or vice versa.) Masking was not used during group allocation or data collection. This ensured that NpHR and YFP mice were interleaved during the experimental sessions and counterbalanced in different behavioral chambers. Masking was a central part of the behavioral analysis: the CNN scores freezing the same regardless of the mouse's experimental group.

Medial or lateral VTA dopamine neurons were inhibited using continuous 594 nm laser illumination (~6 mW, Cobalt) during tone offset or tone onset during extinction (*Figures 5E* and *6A*). Tone onset inhibition lasted for 6 s and started concurrent to the onset of the tone. Tone offset inhibition also lasted 6 s and started 1 s prior to the ending of the tone and lasted for 5 s after the tone offset. Video of the behavior was acquired as normal, with the addition of a 400–650 nm filter (Kentek) in front of the camera lens to prevent laser light from appearing in the video.

Prior to or following fear extinction, the same mice underwent the real time place preference (RTPP) assay with the same laser intensity (*Figure 5—figure supplement 1*). For medial VTA tone off and lateral VTA tone on cohorts, an equal number of mice received RTPP before versus after fear conditioning and extinction. For lateral VTA tone off cohorts, all mice received RTPP after the fear conditioning and extinction assay.

No explicit power analysis was used to determine sample size for fiber photometry experiments. We decided the sample size (n is approximately 10 for each group) based on what is typically seen in the literature. The sample size of approximately N of 10 per group are biological replicates, where biological replicate means an individual mouse. For two of the three optogenetic cohorts, two different rounds of mice were run on separate occasions and data was combined for the full cohort. For the third cohort, all mice were run at once. All mice had at least one fiber in the lateral VTA (from histology images) and were included in statistical analysis. No outliers were discarded. From all cohorts, 1 NpHR mouse was removed from fear extinction analysis because the optic fiber connection slipped off during extinction day 3.

For the data in *Figure 5*, *Figure 6* and *Figure 5—figure supplement 3*, a 2-factor ANOVA with group (NpHR or YFP) and tone number (repeated measure: 1 to 63 extinction tones) as factors was used to predict either (1) freezing during the tone (calculated as mean freezing for 1 to 19 s of the tone), (2) freezing during the 6 s laser offset inhibition period (calculated as the mean freezing for 1 s prior to tone offset through 5 s after tone offset), (3) freezing 6 s post inhibition period (calculated as mean freezing 6–11 s after tone offset) or (4) freezing for 30 s during the ITI (calculated as 12–41 s after tone offset). For the data in *Figure 5—figure supplement 1B–E*, a 2-factor ANOVA with group (NpHR or YFP) and subregion (medial VTA or lateral VTA) was used to determine significance.

## Correlation between GCaMP6f and freezing

No explicit power analysis was used to determine sample size for fiber photometry experiments. We decided the sample size (N is approximately 10 for each group) based on what is typically seen in the literature. No outliers were discarded. For each fiber photometry cohort, 1–4 mice were run on each occasion and all data was aggregated to create the cohorts for the paper. For trial-by-trial analysis, we calculated the Pearson correlation coefficient between GCaMP6f and freezing during extinction for each mouse. Since there were 63 total extinction trials across three extinction days, each mouse had 63 values for mean freezing during the tone, correlated with 63 values for mean GCaMP6f fluorescence for 5 s after tone onset or 5 s after tone offset. We next gathered the correlation coefficients from each region (medial or lateral VTA) and GCaMP6f time point (5 s after tone onset or tone offset) and used a one sample t-test to determine if they significantly differed from 0 (*Figure 3A and D*). For across animal analysis, we used Pearson's correlation coefficient and resulting p-value to find the correlation between mean freezing per mouse with mean GCaMP6f (either 5

s after tone onset or after tone offset) per mouse, we drew the best fit line using estimates from a generalized linear regression model (*Figure 3B,C,E,F*).

## Histology

To confirm the location of viral targeting and optical fiber implant, mice were anesthetized with Euthasol (.1 mL/mouse), perfused with 10 mL of phosphate buffered saline (PBS) followed by 10 mL of 4% paraformaldehyde (PFA) in PBS, their brains extracted, post-fixed with 4% PFA for 24 hrs, then stored in a 30% sucrose in PBS solution for at least 24 hrs before slicing. Brains were sliced coronally at 40 μm, and relevant VTA slices were stained for tyrosine hydroxylase (TH), a marker for dopamine neurons (primary antibody: Chicken Anti-TH, Aves Labs; secondary antibody: Alexa Fluor 647 Donkey Anti-Chicken IgG, Jackson ImmunoResearch) and GFP, to enhance viral expression (primary antibody: GFP Recombinant Rabbit Monoclonal Antibody, Thermo Fisher Scientific; secondary antibody: Alexa Fluor 488 Donkey Anti-Rabbit IgG, Life Technologies). Slices were then mounted with Fluoromount-G with DAPI (Thermo Fisher Scientific) to determine the location of cell nuclei.

## Ex vivo electrophysiology recordings to relate optogenetic inhibition with decreased currents and spiking activity

To confirm optogenetic inhibition of VTA DA cells, we performed ex vivo electrophysiology in Dat-Cre mice. Coronal slices containing the VTA were prepared from 3 to 4 month old male Dat-Cre mice approximately 4 weeks after injecting with AAV5-DIO-NpHR-eYFP virus. Mice were deeply anaesthetized with an intraperitoneal injection of euthasol (0.06 ml per 30 g) and decapitated. After extraction, the brain was immersed in ice-cold carbogenated N-methyl-D-glucamine (NMDG) artificial cerebrospinal fluid (ACSF) (92 mM NMDG, 2.5 mM KCl, 1.25 mM NaH$_2$PO$_4$, 30 mM NaHCO$_3$, 20 mM HEPES, 25 mM glucose, 2 mM thiourea, 5 mM Na-ascorbate, 3 mM Na-pyruvate, 0.5 mM CaCl$_2$·4H$_2$O, 10 mM MgSO$_4$·7H$_2$O and 12 mM N-acetyl-L-cysteine) for 2 min. Afterwards, coronal slices (300 μm) were sectioned using a vibratome (VT1200s, Leica) and then incubated in NMDG ACSF at 34°C for 15 min. Slices were then transferred into a holding solution of HEPES ACSF (92 mM NaCl, 2.5 mM KCl, 1.25 mM NaH$_2$PO$_4$, 30 mM NaHCO$_3$, 20 mM HEPES, 25 mM glucose, 2 mM thiourea, 5 mM Na-ascorbate, 3 mM Na-pyruvate, 2 mM CaCl2·4H$_2$O, 2 mM MgSO$_4$·7H$_2$O and 12 mM N-acetyl-L-cysteine, bubbled at room temperature with 95% O$_2$, 5% CO$_2$) for at least 45 min until recordings were performed. Whole cell recordings were performed using a Multiclamp 700B (Molecular Devices, Sunnyvale, CA) using pipettes with a resistance of 3–5 MΩ filled with a potassium-based internal solution containing 120 mM potassium gluconate, 0.2 mM EGTA, 10 mM HEPES, 5 mM NaCl, 1 mM MgCl$_2$, 2 mM Mg-ATP and 0.3 mM NA-GTP, with the pH adjusted to 7.2 with KOH. DA neurons in the VTA were identified for recordings based on YFP expression. Photostimulation parameters were 560 nm and 5 mW/mm$^2$. Neurons were held at −70 mV during photocurrent measurements. To confirm the ability of photocurrents to eliminate action potentials, action potentials were induced by a positive current injection (120 pA, 100 ms pulse duration, 5 Hz).

## CNN

We introduce an open-source pipeline to automate freezing analysis of behavioral videos. The main motivation and advantage of this pipeline is automation of freezing analysis, even when mice are tethered to neural headgear – our analysis does not confuse headgear movement for mouse movement. Drawbacks to this technique include necessity for manual scoring to train the network (it took approximately seven hours to score the 33,000 images we used to train each network), and different networks must be trained for different backgrounds and neural headgear. Additionally, having a graphics processing unit (GPU) greatly speeds up network training (1–2 hrs to train 200 epochs on a 320 nVidia P100 GPU vs. 1 week on a 32 GB RAM, Intel Core i7 CPU) and using the network for analysis (10 min to run 25,000 frames on a 320 nVidia P100 GPU vs. 1 hr on a 32 GB RAM CPU).

Training and using the network involved: (1) Human labeling of freezing behavior, (2) Generating images for CNN input, (3) Training the CNN, (4) Assessing CNN accuracy and precision, and (5) Using network for analysis. Custom MATLAB software was developed for parts (1 - 2), and custom Python software was developed for parts (3 - 5). Code for all steps is available on GitHub: https://github.com/neurocaience/deepfreeze/ (*Cai et al., 2020*; copy archived at https://github.com/elifesciences-publications/deepfreeze).

1. Human Labeling of Freezing behavior

   First, behavioral videos were broken down into individual frames. Next, random pairs of consecutive frames were selected for human labeling as '1' ('freeze' - no mouse movement occurred between frames) or '0' ('no freeze' - mouse movement occurred between frames). Scored images were saved and processed in step (2). To ensure that all possible mouse movements are represented, we recommend using behavioral videos from random time points of at least 7 to 20 mice for each context. In particular, the fear conditioning context requires 15–20 mice due to low variability of movements within mice resulting from highly prevalent freezing. Mice in the fear extinction context are more apt to moving around, thus 5–7 mice are sufficient to capture this variability. Each network needs 30–40 k labels, or approximately 7 hrs of human labeling. The network trains best with roughly equal amounts of 'freeze' and 'no freeze' data.

2. Generating images for CNN input

   Pairs of consecutive frames used for human labeling were condensed into a 'difference image':

   $D_{i,j}$. This difference image reflects the threshold-normalized absolute value of the pixel intensity difference between consecutive images.

   $$D_{i,j} = uint8\left[ max\left( \frac{\Delta P - \mu(\Delta P)}{\sigma(\Delta P)} , \theta \right) \cdot \frac{255}{\theta} \right]$$

   Where:

   $D_{i,j}$ is the difference image

   $i, j \ni 1, 2, ...N$ where $N$ is the number of pixels along each axis of the image

   $\Delta P = abs\left( P_{i,j,t} - P_{i,j,(t+1)} \right)$ where $P$ is an individual frame and $t$ is the frame number in the video

   μ is the mean function

   σ is the standard deviation function

   θ is the threshold value, the same threshold is used for all $D_{i,j}$ calculations. In our case, we used $\theta = 15$, set as approximately the top 0.05% percentile of $\Delta P$.

   $\frac{255}{\theta}$ is a scale factor that ensures the pixel value is at a maximum of 255 (for grayscale images, pixel intensity values range from 0 to 255).

   $uint8$ is a function that transforms a floating point number into an 8-bit integer between 0 to 255, so the $D_{i,j}$ can be saved into a. png file.

3. Training the CNN

   Next, the difference images, along with their respective human labels, were split across mice into train and test sets, where one mouse's images were either in the training or test set, but not both. This was analogous to K-fold cross validation, but where the data are partitioned by mouse rather than randomly. This ensured that similarity of behavior within a mouse did not confound accuracy of the test dataset.

   The training set was then used to train a CNN (ResNet18 *He et al., 2016*, pretrained on ImageNet, batch size = 128, learning rate = 0.001, Adam optimization *Kingma and Ba, 2014*, random rotation and flip) over the course of 200 epochs; the CNN weight matrix of each epoch was saved for analysis and network selection in the next section. The CNN aimed to minimize cross entropy loss (a form of negative log likelihood) using the PyTorch module (*Paszke et al., 2019*).

4. Assessing CNN Accuracy and Precision

   To ascertain which CNN epoch to use as the final classifier, we examined its test loss, accuracy, false positive rate and false negative rate. We accepted epochs when test loss plateaued, accuracy was above 90%, and false positive rate (FPR) and false negative rate (FNR) were below or around 10%. The exact epoch varried with each network, but ranged from 20–40 epochs. The CNN weight matrix from the appropriate epoch was then chosen as the classifier for behavioral analysis. Below are the methods used for calculating false positives, false negatives, true positives, true negatives, false positive rate and false negative rate:

   False positives (FP): Number of frames that were classified as 'freeze', but labeled 'no freeze'

   False negative (FN): Number of frames that were classified as 'no freeze', but labeled 'freeze'

   True positives (TP): Number of frames that were classified as 'freeze' and labeled 'freeze'

   True negatives (TN): Number of frames that were classified as 'no freeze' and labeled 'no freeze'

   False positive rate (FPR): $\frac{FP}{FP + TN}$

False negative rate (FNR): $\frac{FN}{FN + TP}$

5. Using Network to Classify Freezing

All behavioral videos are broken down into individual frames, which were then formatted into 'difference images'. These difference images were fed as input into a trained CNN, which then predicted an output of either '1' or '0', representing a 'freeze' or 'no freeze'. Post-processing involved averaging the predicted label for each frame across each second of video, to obtain %freeze/s. Our videos had a frame rate of 11.2 Hz.

We trained separate networks for each of our four experimental contexts (fear conditioning vs fear extinction; fiber photometry vs. optogenetics), and used the above steps (1 - 5) to identify the frames in which mice were freezing during our behavioral videos.

## Acknowledgements

We thank H Wang for assistance with labeling behavioral videos, R Witten for CNN coding proof of concepts and discussions, M Murugan, V Corbit and R Cho for comments on this paper, Y Niv for advice on this project, as well as other members of the Witten lab for their feedback, advice and support. This research was funded by NYSCF, Pew, McKnight, NARSAD, ARO W911NF1710554, NIH U19 NS104648-01, NIH DP2 DA035149-01, NIH 5R01MH106689-02, NIH 5R01DA047869 to IBW, NIH T32 MH065214 to LXC and NIH F32 MH112320-03 to JMC. IBW is a New York Stem Cell Foundation—Robertson Investigator.

## Additional information

### Funding

| Funder | Grant reference number | Author |
| --- | --- | --- |
| National Institutes of Health | T32MH065214 | Lili X Cai |
| New York Stem Cell Foundation | | Ilana B Witten |
| Army Research Office | W911NF1710554 | Ilana B Witten |
| National Institutes of Health | 1R01MH106689-01A1 | Ilana B Witten |
| McKnight Foundation | | Ilana B Witten |
| National Alliance for Research on Schizophrenia and Depression | | Ilana B Witten |
| National Institutes of Health | U19NS104648-01 | Ilana B Witten |
| National Institutes of Health | DP2 DA035149-01 | Ilana B Witten |
| National Institutes of Health | 5R01MH106689-02 | Ilana B Witten |
| Pew Charitable Trusts | | Ilana B Witten |
| National Institutes of Health | F32MH112320-03 | Julia M Cox |
| National Institutes of Health | 5R01DA047869 | Ilana B Witten |
| New York Stem Cell Foundation | Robertson Investigator | Ilana B Witten |

The funders had no role in study design, data collection and interpretation, or the decision to submit the work for publication.

### Author contributions

Lili X Cai, Conceptualization, Data curation, Software, Formal analysis, Supervision, Validation, Investigation, Visualization, Methodology, Writing - original draft, Writing - review and editing; Katherine Pizano, Cameron L Hayes, Weston T Fleming, Sebastian Holt, Investigation; Gregory W Gundersen, Software; Julia M Cox, Methodology; Ilana B Witten, Conceptualization, Supervision, Funding acquisition, Visualization, Project administration, Writing - review and editing

## Author ORCIDs
Lili X Cai ⓘ https://orcid.org/0000-0003-3260-2367
Cameron L Hayes ⓘ http://orcid.org/0000-0002-0388-5807
Ilana B Witten ⓘ https://orcid.org/0000-0003-0548-2160

## Ethics
Animal experimentation: All experiments followed guidelines established by the National Institutes of Health and reviewed by Princeton University Institutional Animals Care and Use Committee (IACUC protocol 1876-18).

## Decision letter and Author response
Decision letter https://doi.org/10.7554/eLife.54936.sa1
Author response https://doi.org/10.7554/eLife.54936.sa2

## Additional files

### Supplementary files
• Transparent reporting form

### Data availability
All neural recordings, behavioral data and code to reproduce figures is at: https://github.com/neuro-caience/DopamineAndFear (copy archived at https://github.com/elifesciences-publications/DopamineAndFear). Code for human labeling and the CNN pipeline is at https://github.com/neurocaience/deepfreeze (copy archived at https://github.com/elifesciences-publications/deepfreeze).

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
