## [Decision Letter]

**Acceptance summary:**

This study shows a novel type of diversity within dopamine neurons in the ventral tegmental area (VTA). The use of fear extinction allowed the authors to reveal novel aspects of dopamine signals, in particular, salience-related signals in the lateral VTA contrasting with reward prediction error-like signals observed in the medial VTA. Optogenetic inhibition experiments further supports functional differences between dopamine neurons in the medial and lateral VTA. These results together provide evidence that extends the notion of heterogeneity in dopamine neurons in a significant manner.

**Decision letter after peer review:**

Thank you for submitting your article "Distinct signals in medial and lateral VTA dopamine neurons modulate fear extinction at different times" for consideration by *eLife*. Your article has been reviewed by three peer reviewers, including Naoshige Uchida as the Reviewing Editor and Reviewer #1, and the evaluation has been overseen by Kate Wassum as the Senior Editor. The following individual involved in review of your submission has agreed to reveal their identity: Satoshi Ikemoto (Reviewer #3).

The reviewers have discussed the reviews with one another and the Reviewing Editor has drafted this decision to help you prepare a revised submission.

Summary

The authors examine the role of ventral tegmental area dopamine neurons (VTA-DA) in the extinction of fear learning. Using fiber photometry they demonstrate that medial VTA-DA neuronal populations are inhibited by aversive predictive auditory tones and excited when an expected aversive shock is omitted during fear extinction learning. They also report that lateral VTA-DA neuron populations respond to aversive predictive tones and to shock omission. Using correlation analysis of neural activity on a trial-by-trial and across animal basis, they then show that the tone evoked activity during fear extinction in medial VTA-DA cells is negatively correlated with extinction learning while the shock omission related activity is positively correlated with freezing. Contrasting with this, lateral VTA-DA neuronal activity is positively correlated with the predictive tone. Finally, they use optogenetics to show that inhibition of medial VTA-DA neurons during shock omission reduces extinction while inhibiting lateral VTA-DA neurons only has an effect during the tone onset.

This is a timely study because there are increasing interests in diversity of DA neurons. Recent studies have shown fear extinction is regulated by DA neurons (e.g. Luo et al., 2018; Salinas-Hernandez et al., 2018), but the results of the current study extend these recent studies and provide important information on distinct neuronal activation in medial and lateral VTA-DA cells. The different effects of optogenetic inhibition between these cell populations are also potentially very important. The study is conducted with sound methods and care. The manuscript is written clearly and the results are presented well generally.

While all the reviewers thought that this manuscript contains important results, it was pointed out that some findings of the present work overlaps with previous studies (Luo et al., 2018; and Salinas-Hernandez et al., 2018). This does not necessarily reduce the significance of the current work, but the authors should clarify the novelty of the current work and the difference from the previous studies, while giving more proper credits to previous studies. Reviewers also found various technical and interpretational issues that need to be addressed. We therefore would like to invite the authors to respond to these concerns before publication of this work in *eLife*.

Essential revisions:

1) The authors interpret the results as though they are specifically manipulating the medial and lateral VTA (highlighted in the title, as well). However, based on the fiber locations for photometry and optogenetics, some of the fibers for the "lateral VTA" appear to encompass the substantial nigra pars compacta (SNc). While this does not necessarily reduce the significance of the study, the results need to be reframed to clarify that it remains unclear whether the activity and optogenetic effects observed in this study come from lateral VTA or SNc.

2) Previous studies demonstrated that VTA-DA neuronal activity and the release of DA in nucleus accumbens (NAc) increases when anticipated shocks are omitted (Badrinarayan et al., 2012; Salinas-Hernández et al., 2018). Furthermore, prior work reported that inhibition of VTA-DA cells or medially situated VTA-DA projections to the medial NAc shell, but not more laterally positioned VTA-DA projections to NAc core, impairs extinction learning (Salinas-Hernández et al., 2018; Luo et al., 2018). The present study replicates some of these findings and extends them by imaging from medial and lateral VTA-DA cell groups and showing distinct coding features (RPE vs. Salience) as well as trial-by-trial correlations of activity and behavior. Furthermore, they report that inhibition of lateral VTA-DA neurons during the cue slows extinction. While these are interesting findings, they are a bit incremental and their importance could be better framed in the context of the previous studies.

3) In Figure 4E the authors show that medial VTA-DA neurons are excited at tone offset and that this decreases over the course of multiple days of extinction. To rule out the possibility that this is simply a response to the cessation of the tone itself which decays over days and not due to a prediction error set up by learning they should analyze and show the response to the tone before fear conditioning (habituation).

4) The authors argue that medial VTA-DA neurons encode a reward prediction error, but during conditioning the VTA-DA neurons exhibit a large response to the electrical shocks (Figure 3—figure supplement 1G). This suggests that the cell population they are recording from do not encode simply reward prediction error. They are using a fiber photometry approach which captures what could be heterogeneous medial VTA cell populations (some of which project to mPFC are may be aversive coding) which could account for this. Whatever the resolution, the authors should discuss this apparent discrepancy with their working model in more detail.

5) As the authors mention, (motivational) salience can be defined as "an unsigned prediction error" or "the absolute value of RPE". The data appear to be preliminary to conclude that lateral VTA-DA neurons signal salience. First, as discussed in the point #1, whether the excitation at the CS onset and the termination of the CS comes from VTA or SNc. Second, the onset and the end of CS are often the time at which a sudden change in behavior occurs (e.g. see Figure 3D, freeze %). It is possible that the observed activation reflects a switch in behavior much like movement onset signals observed in previous studies. Third, the excitation at the end of CS is rather small. To strengthen the conclusion, it would be preferable to observe whether they are strongly activated by reward. Also, do the authors observe excitation during the US (footshock) during conditioning? Fourth, optogenetic inhibition of lateral VTA-DA neurons caused place aversion and decreased velocity. These results do not necessarily align with the idea that lateral VTA-DA neurons signal salience.

6) Potentially the most important result in this study is slowing of fear extinction by optogenetic inhibition of lateral VTA-DA neurons at CS onset. However, this result is underdeveloped and appear preliminary. First, the result appears mainly due to increased freezing in the last half of Extinction Day 3. In the control experiment, the authors showed that optogenetic inhibition decreases the velocity of the animals' movement. It is unclear whether the authors can separate slowing of movement (which may increase "apparent" freezing) and a direct effect on extinction (i.e. a process of reducing a learned behavior). Second, the effect is relatively small. Perhaps, Figure S5 shows the effect more clearly than Figure 5 does (Please show this and a quantitative summary in the main figure). But still, the two conditions intersect. Third, there is significant decrease in the baseline (before CS) on Extinction Day 1 and 2. It is unclear whether this affected the data analysis. Overall, although this is one of the most important results of this study, the data are not particularly strong.

---

## [Author Response]

Essential revisions:1) The authors interpret the results as though they are specifically manipulating the medial and lateral VTA (highlighted in the title, as well). However, based on the fiber locations for photometry and optogenetics, some of the fibers for the "lateral VTA" appear to encompass the substantial nigra pars compacta (SNc). While this does not necessarily reduce the significance of the study, the results need to be reframed to clarify that it remains unclear whether the activity and optogenetic effects observed in this study come from lateral VTA or SNc.

Thank you for your important comment, and the detailed attention to the histology of fiber placements. We agree that light at the fiber tip could indeed overlap between lateral VTA and SNc. However, we would like to point out that the effects reported in this study appear more likely to arise from lateral VTA than SNc. This conclusion is due both to a detailed examination of our fiber placements and virus spread, and also direct comparisons of data that we collected in SNc rather than lateral VTA.

Regarding lateral VTA fiber photometry placements, we note that all our lateral VTA placements are within the boundaries of lateral VTA, and do not overlap with SNc. (We have now excluded the animal with a fiber that is closest to SNc, and mention this exclusion in the Materials and methods.) For lateral VTA optogenetic experiments, all mice either had much stronger virus expression in the VTA than SNc, and/or or the placement was clearly in VTA.

To directly compare SNc fiber photometry signals to those measured in lateral VTA, we recorded from 2 additional mice using SNc coordinates (-2.9 posterior, -1.25 lateral, -4.2 ventral relative to Bregma; Figure 3—figure supplement 3). We saw results that qualitatively differed from lateral VTA: during habituation and conditioning, SNc DA neuron activity responded to the shock but not to the cue, while lateral VTA DA responds to both shock and cue (Figure 3—figure supplement 3). During extinction, SNc DA neurons displayed minimal increase in activity at cue onset, while lateral VTA showed robust increase at that time (Figure 3F). Furthermore, unlike lateral VTA, SNc did not show increased activity at the cue offset. Thus, it seems very unlikely that our lateral VTA data could be fully explained by SNc recordings.

Furthermore, we conducted optogenetic inhibition experiments in mice with fibers and virus targeting the SNc, and inhibited SNc DA neurons during the extinction tone offset (Figure 5—figure supplement 3). Similar to inhibiting lateral VTA DA neurons at tone offset, inhibiting SNc DA neurons also did not have an effect on tone freezing. However, there seemed to be an effect of decreased freezing during the ITI ( 2-way ANOVA with group and trial, where trial is a repeated measure, to predict freezing during 30 s of the inter-trial interval: group effect: F_(1,62)_ = 4.980, p = 0.047) , which we did not observe with lateral VTA cue offset inhibition. This suggests that lateral VTA and SNc inhibition during the cue offset may not have identical consequences, although these differences are not as definitive as the fiber photometry results.

In summary, we include the SNc data described above in our paper as supplemental figures, as it suggests, especially in the case of fiber photometry, that our lateral VTA results cannot fully be explained by SNc.

2) Previous studies demonstrated that VTA-DA neuronal activity and the release of DA in nucleus accumbens (NAc) increases when anticipated shocks are omitted (Badrinarayan et al., 2012; Salinas-Hernández et al., 2018). Furthermore, prior work reported that inhibition of VTA-DA cells or medially situated VTA-DA projections to the medial NAc shell, but not more laterally positioned VTA-DA projections to NAc core, impairs extinction learning (Salinas-Hernández et al., 2018; Luo et al., 2018). The present study replicates some of these findings and extends them by imaging from medial and lateral VTA-DA cell groups and showing distinct coding features (RPE vs. Salience) as well as trial-by-trial correlations of activity and behavior. Furthermore, they report that inhibition of lateral VTA-DA neurons during the cue slows extinction. While these are interesting findings, they are a bit incremental and their importance could be better framed in the context of the previous studies.

Thank you for this comment. We fully agree that placing our results in the context of other studies is essential. For this reason, we had referenced Badrinarayan et al., 2012 and Salinas-Hernández et al., 2018 in the Introduction and Discussion:

Introduction:

“In particular, VTA DA neurons were shown to represent RPE-like signals during fear extinction, in that they display elevated activity when the shock is omitted at the offset of the cue, signaling better-than-expected outcome ( Salinas-Hernández et al., 2018; Badrinarayan et al., 2012; Jo, Heymann and Zweifel, 2018). ”

Discussion:

“Our manipulations of medial VTA DA neurons produced changes in extinction learning that were consistent with the presence of an RPE-like signal in that subregion. Specifically, the burst of DA at the omission of the expected shock contributed causally to extinction learning, a finding which aligned with previous work ( Salinas-Hernández et al., 2018; Luo et al., 2018).”

For the resubmission, we added the following sentences:

1) In the Results section: “This was consistent with previous work that showed inhibiting medial VTA during cue offset impairs extinction learning (Luo et al., 2018; Salinas-Hernández et al., 2018).”

2) In the Discussion section: “We found that inhibition of the lateral VTA at the tone offset did not influence extinction learning of the preceding tone. This was consistent with previous work showing inhibition of a subpopulation of VTA DA neurons with projections to NAc Core did not affect extinction (Luo et al., 2018)”.

3) In Figure 4E the authors show that medial VTA-DA neurons are excited at tone offset and that this decreases over the course of multiple days of extinction. To rule out the possibility that this is simply a response to the cessation of the tone itself which decays over days and not due to a prediction error set up by learning they should analyze and show the response to the tone before fear conditioning (habituation).

We appreciate this comment and point to Figure 3—figure supplement 1G, where we show that during habituation, medial VTA DA activity at the tone offset is not different from baseline.

4) The authors argue that medial VTA-DA neurons encode a reward prediction error, but during conditioning the VTA-DA neurons exhibit a large response to the electrical shocks (Figure 3—figure supplement 1G). This suggests that the cell population they are recording from do not encode simply reward prediction error. They are using a fiber photometry approach which captures what could be heterogeneous medial VTA cell populations (some of which project to mPFC are may be aversive coding) which could account for this. Whatever the resolution, the authors should discuss this apparent discrepancy with their working model in more detail.

We agree with the reviewers that medial VTA DA neurons do not simply encode RPE during all behaviors. In both the Results and Discussion, we specify that the activity seems to be consistent with the RPE framework during fear extinction, while neither RPE nor salience could fully explain neural activity in either subregion during conditioning. We agree this might be due to heterogeneity on the single cell level that we do not pick up with fiber photometry.

To address your points, in the Discussion section we write: “In fact, we found that during fear conditioning, medial and lateral VTA shared similar activity profiles, which may reflect heterogeneity in single neuron activity that is not captured with fiber photometry.”

5) As the authors mention, (motivational) salience can be defined as "an unsigned prediction error" or "the absolute value of RPE". The data appear to be preliminary to conclude that lateral VTA-DA neurons signal salience. First, as discussed in the point #1, whether the excitation at the CS onset and the termination of the CS comes from VTA or SNc.

We agree this is an important point. We point the reviewer to our response to comment #1, where we show a different fiber photometry profile in SNc versus lateral VTA.

Second, the onset and the end of CS are often the time at which a sudden change in behavior occurs (e.g. see Figure 3D, freeze %). It is possible that the observed activation reflects a switch in behavior much like movement onset signals observed in previous studies.

We agree with the reviewers that this is a good point. To help address this question, we now include a figure that displays the cross-correlation between lateral VTA DA neuron activation (GCaMP6f fluorescence) and freezing during habituation, conditioning, and extinction (Author response image 1).

**Author response image 1. respfig1:** Crosscorrelation between medial and lateral VTA dopamine neuron activity and freezing behavior. (**A**) Crosscorrelation between medial VTA dopamine neuron activity and freezing behavior during different experiment conditions: habituation (top plot), fear conditioning (middle plot), and fear extinction (bottom plot). For each plot, red line is the mean cross-correlation across mice, pink shading denotes SEM. Grey lines denote cross-correlation of an individual mouse for each experiment condition (n = 10 mice for habituation and fear conditioning sessions, n = 10 mice with 3 extinction days per mouse for fear extinction session). (**B**) Same as A, but for Lateral VTA, using blue instead of red line and shading to represent mean crosscorrelation across mice (n = 11 mice).

We observed a positive correlation in lateral VTA between freezing and GCaMP6f during habituation, which is in the correct direction to explain the salience-like signal during extinction at the tone onset, when we observed increased freezing and increased GCaMP6f fluorescence. However, this positive correlation does not explain the salience-like signal during extinction at the tone offset, when we observed decreased freezing and increased GCaMP6f fluorescence. This suggests decoupling between the salience-like signal and a direct relationship between freezing and GCaMP6f. Note also that the correlation between GCaMP6f and freezing was stronger during habituation than extinction in lateral VTA, even though the salience-like signal was much stronger during extinction. This further suggests that the correlation between freezing and GCaMP6f is not sufficient to explain the salience-like effect.

Third, the excitation at the end of CS is rather small. To strengthen the conclusion, it would be preferable to observe whether they are strongly activated by reward.

We agree with the reviewer that confirming that lateral VTA neurons are also activated by reward would strengthen the conclusion that lateral VTA reflects salience. We have previously demonstrated reward responses throughout the medial/lateral extent of the VTA with 2-photon imaging (Engelhard et al., 2019). To check on this result with fiber photometry, we presented a group of mice (n = 3) with uncued random rewards (sweetened condensed milk) in an operant chamber (Figure 3—figure supplement 2). We observed increased activity in lateral VTA dopamine neurons to both reward consumption (which we aligned with the first lick onset after reward delivery) and shock (Figure 3—figure supplement 1C). We note that increased activity to reward consumption started approximately 0.5 s before lick onset, which could be due to reward approach as others have reported (Hamid et al., 2016; Howe et al., 2013; Day et al., 2007). We conducted the same experiment in a mouse with a fiber located above medial VTA DA neurons and saw similar results (Figure 3—figure supplement 2).

Also, do the authors observe excitation during the US (footshock) during conditioning?

Yes, we reported excitation during the US (footshock) during conditioning (Figure 3—figure supplement 1H).

Fourth, optogenetic inhibition of lateral VTA-DA neurons caused place aversion and decreased velocity. These results do not necessarily align with the idea that lateral VTA-DA neurons signal salience.

Thank you for pointing this out. We now emphasize in the Discussion that our findings suggest lateral VTA-DA neurons signal a salience-like signal *during* *fear extinction,* but our conditioning data and other datasets suggest additional roles or further complexity to this signal during other behaviors.

6) Potentially the most important result in this study is slowing of fear extinction by optogenetic inhibition of lateral VTA-DA neurons at CS onset. However, this result is underdeveloped and appear preliminary. First, the result appears mainly due to increased freezing in the last half of Extinction Day 3. In the control experiment, the authors showed that optogenetic inhibition decreases the velocity of the animals' movement. It is unclear whether the authors can separate slowing of movement (which may increase "apparent" freezing) and a direct effect on extinction (i.e. a process of reducing a learned behavior).

This is a very important point. Since VTA inhibition does decrease mobility in our real time place aversion (RTPA) experiment, we cannot rule out a direct effect of lateral VTA inhibition at the tone onset on impacting the expression of freezing rather than the learning process itself. However, the fact that we do observe a significant interaction between NpHR/YFP group and extinction trial suggests an effect on learning. Relatedly, as you point out, the significant difference with the post-hoc test comparing groups only appears on the last day, again consistent with a gradual learned effect (Figure 6). However, we do agree that with this data we cannot rule out an effect on expression rather than learning, and we bring up this caveat in the Discussion.

Second, the effect is relatively small. Perhaps, Figure S5 shows the effect more clearly than Figure 5 does (Please show this and a quantitative summary in the main figure). But still, the two conditions intersect.

We agree this effect is relatively small, although it is statistically significant. We include the panel in question from Figure S5 in Figure 6, as you recommend. In addition to reporting a significant interaction between group and extinction trial in predicting freezing during the cue (Figure 6D; 2-way mixed ANOVA where group and tone number are factors, and freezing during the tone the dependent variable. Group effect: F_(1,62)_ = 0.014, p = 0.908; group × tone number interaction: F_(1,62)_ = 2.217 p < 10^-6^ )^^, we now also include the post-hoc tests on individual trials comparing groups, which is significant for some of the trials on day 3, but not the other days (Figure 6C). We agree it is better to have all the visualizations in the main figure. Since the effect is small, we decided to edit some of the language in the Discussion accordingly. Specifically, we write: “… the effect on behavior [of lateral VTA manipulation] was relatively subtle.”

Third, there is significant decrease in the baseline (before CS) on Extinction Day 1 and 2. It is unclear whether this affected the data analysis. Overall, although this is one of the most important results of this study, the data are not particularly strong.

It is true that lateral VTA inhibition affects freezing during the ITI (which you are referring to as baseline freezing), not just during the CS. For simplicity, we had not discussed that in our original submission. We now report that this difference in ITI freezing is indeed significant (2- factor ANOVA with group and tone number as factors, tone number being a repeated measure: Group effect: F_(1,62)_ = 6.477, p = 0.019; group × tone number interaction: F_(1,62)_ = 1.562 p = 0.004). Interestingly, the effect on the ITI opposes the effect on the tone, in that initially there is less freezing in the NpHR group during the ITI, and later there is more freezing in the NpHR group during the tone. A possible interpretation is that lateral VTA DA at tone onset promotes freezing acutely (thus inhibition decreases freezing acutely during the ITI), but over time it contributes to extinction learning (and increases freezing during the tone).